# Amino Acid Profiles and Nutritional Evaluation of Fresh Sweet–Waxy Corn from Three Different Regions of China

**DOI:** 10.3390/nu14193887

**Published:** 2022-09-20

**Authors:** Ziqi Li, Tiannuo Hong, Guohui Shen, Yanting Gu, Yanzhi Guo, Juan Han

**Affiliations:** 1Institute of Food and Nutrition Development, Ministry of Agriculture and Rural Affairs, Beijing 100081, China; 2Laboratory of Safety & Nutritional Function Risk Assessment for Agricultural Products of China Ministry of Agriculture and Rural Affairs, Beijing 100081, China

**Keywords:** fresh corn, amino acid profiles, amino acids evaluation, nutritional evaluation, factor analysis, PCA

## Abstract

This study conducted a comparative analysis of the amino acid compositions of Chinese Huangnuo 9 fresh sweet–waxy corn from three different provinces in China—Inner Mongolia, Jilin, and Heilongjiang Province. Moreover, we established a nutritive evaluation system based on amino acid profiles to evaluate, compare, and rank the fresh sweet–waxy corn planted in different regions. A total of 17 amino acids were quantified, and the amino acid composition of fresh sweet–waxy corn was analyzed and evaluated. The amino acid quality was determined by the amino acid pattern spectrum, chemical evaluations (including CS, AAS, EAAI, BV, U(a,u), NI, F, predict PER, and PDCAAS), flavor evaluation, amino acid matching degree evaluation, and the results of the factor analysis. The results showed that the protein content of fresh corn 1–1 from Inner Mongolia was the highest (40.26 ± 0.35 mg/g), but the factor analysis results, digestion, and absorption efficiency of fresh corn 1–2 were the best. The amino acid profile of fresh corn 1–1 was closest to each evaluation’s model spectrum. The results of the diversity evaluations in fresh corn 3–2 were the best, and fresh corn 3–3 had the most essential amino acid content. A total of 17 amino acids in fresh corn were divided into three principal component factor analyses: functional principal components (Leu, Pro, Glu, His, Ile, Ser, Met, Val, Tyr, Thr), regulatory principal components (Lys, Gly, Ala, Asp, Arg, Trp), and protection principal components (Phe). The scores of the three principal components and the comprehensive score in fresh corn 1–2 were all the highest, followed by 3–3 and 1–1. The amino acid nutritional values of fresh corn 1–2 were the highest in 12 samples.

## 1. Introduction

Corn picked from the late milk stage or early dough stage is called fresh corn. Fresh corn is usually divided into sweet corn, waxy corn, and sweet–waxy corn. Sweet corn is native to America and was introduced into China in the 1980s. It is also called fruit corn due to its sweet, crisp, and juicy taste, and it is well received by consumers. Waxy corn is a new variety produced by a mutation of the Wx gene (which controls the synthesis of a particular protein) at locus 59 on the ninth chromosome of maize after it was introduced into China in the 16th century. The mutation reduces the activity of UDP–glucuronosyltransferases (UGTs) in maize and results in the inability of corn to synthesize amylose [1]. Waxy corn has a long planting history in China; moreover, it is widely believed that waxy corn was first discovered in China [2]. Sweet–waxy corn is a hybrid of sweet corn and waxy corn, with two types of seeds on the same ear and a sweet and sticky taste; in recent years, it has been favored by consumers. The planting areas of fresh corn in China are increasing; the total planting areas of fresh corn in China have exceeded 20 million mu (equivalent to 1.3 million hectares). The total output value of the fresh corn industry reached RMB 3.2 billion in 2021 in the Jilin province.

Protein is an essential macronutrient in the human body and has a variety of physiological functions. Amino acids are components of protein; the content and composition of amino acids directly affect the quality of food proteins. Modern nutrition theory research shows that a deficiency or excess of amino acids affects the nutritional value of food [3]. Carbohydrates are responsible for the taste of corn, but amino acids determine the nutritional value of corn. Foods contain complete varieties of essential amino acids (EAAs), and the proportion of amino acids in food corresponds to the proportion of amino acids required by the human body (high nutritional value food protein) [4]. Analyzing the composition and evaluating the quality of amino acids are significant references that guide the dietary nutrition guidelines of a population, promote the classifications of food management, and assist in the development of new food resources [5,6,7].

Food nutrition evaluations are comprehensive analyses of the quantity and quality of food and food nutrients. Plant protein sources are rich in amino acids and are important sources of protein for human beings [8]. The content and composition of amino acids are the main factors that determine the ratios of food protein utilization. Studies have shown that protein and amino acid deficiencies mainly affect people from developing or less developed countries/regions [9]. Chinese residents have sufficient dietary energy levels and nutritional intake, and the demand for protein in the diet has shifted from quantity to quality [10]. In addition to amino acid content, the amino acid composition is also an important index used to evaluate the nutritional quality of plant proteins [11]. The quality of food amino acids mainly refers to the compositions of amino acids in food (including content and type), the similarity between the composition of food amino acids, the pattern spectra of amino acids, the utilization rate of amino acids in organisms, and the suitability of amino acids for digestion, absorption, and synthesis in the human body [12]. Amino acids that makeup proteins can be divided into various types according to different bases, such as EAAs and non–essential amino acids (NEAAs), taste amino acids (including umami, sweet, and bitter amino acids), and medicinal amino acids [13,14]. Essential amino acids can only be obtained from food instead of being synthesized by the body, so the content and composition of EAAs determine the nutritional values of food proteins. There are eight types of essential amino acids for adults; infants have nine types of essential amino acids because they cannot synthesize histidine (His). Most amino acids have taste; they add layers of flavor to food. Glutamic acid (Glu) and aspartic acid (Asp) are umami amino acids; alanine (Ala), glycine (Gly), serine (Ser), proline (Pro), and threonine (Thr) are sweet amino acids; leucine (Leu), isoleucine (Ile), valine (Val), phenylalanine (Phe), methionine (Met), histidine (His), and arginine (Arg) are bitter amino acids [15]. Medicinal amino acids include Glu, Asp, Arg, Gly, Thr, tyrosine (Tyr), Met, Leu, and lysine (Lys); they are essential for maintaining nitrogen balance in the human body [16].

In addition to the analysis of amino acid compositions, the chemical evaluation, pattern spectrum evaluation, and principal component analysis (PCA) of amino acids are also important in the quality evaluation of food amino acids [10,17]. The three evaluation methods will be described in detail in Section 2 (materials and methods).

The purpose of this study was to clarify the amino acid profiles in fresh corn from three different regions of China, and their influences on the nutrition and flavor of fresh corn, to provide ideas and methods for comprehensive nutritional value evaluation of fresh corn, and improve/provide a scientific basis for the market selections of fresh corn. The nutritional value of fresh corn was analyzed to provide ideas to further measure the relationship between amino acids in fresh corn and the bioactive functioning of fresh corn; suggestions for complementary intake of amino acids in other foods are presented based on protein complementation. This study will provide a scientific reference for the classification of the nutritional quality of fresh corn.

The differences between fresh corn and ordinary corn are not only about the different varieties and plucking times but also the discrepancies in nutritive indices, such as amino acids. In terms of varieties, the main nutrients (such as protein, crude fat, starch, etc.) of fresh corn are higher than those of ordinary maize, rice, and flour [18]. At present, the dietary patterns and nutrient intake compositions of consumers are changing. In recent years, consumers have been paying more attention to the nutritional quality of food [12]. People consume nutrients to improve their health, reduce the incidence of disease, and ultimately extend their lives [19]. Analyses and evaluations of nutritional components not only help improve societal nutritional awareness and health levels, but also act as theory bases that governments use to supervise markets, formulate policies, and carry out targeted food nutrition intervention programs [20]. On the other hand, the results of amino acid evaluations can be applied to the grading of agricultural products, providing a theoretical basis for standardizing the market [21].

## 2. Materials and Methods

### 2.1. Reagents

The following instruments and reagents were used in this study: the laboratory homogenizer was purchased from Beyotime (Shanghai, China). The analytical balance was purchased from AsOne (Osaka, Japan). The KDN–1 automatic Kjeldahl apparatus was from INESA Scientific Instrument Co., Ltd. (Shanghai, China). The amino acids analyzer was from Scion in Techcomp (Shanghai, China). Hydrochloric acid (GR), phenol, sodium citrate (GR), sodium hydroxide (GR), copper sulfate (AR), potassium sulfate (AR), sulfuric acid (AR), boric acid (AR), methyl red (AR), methylene blue (AR), sodium hydroxide (AR), and ethanol (95%) were purchased from Sigma–Aldrich (Madrid, Spain). Nitrogen was purchased from Aladdin (Shanghai, China). The standard solution of mixed amino acids and a standard solution of 17 amino acids were from Wako (Osaka, Japan).

### 2.2. Materials and Sample Preparation

Fresh sweet–waxy corn (Huangnuo 9) was collected from the Fuyuanquan Dongda ecological planting base in Wang Aizhao town, Dalad Banner, Erdos City, Inner Mongolia autonomous region, Changchun planting base in Jilin province, and Suihua planting base in Heilongjiang province. The different measuring points were set according to the different planting densities in the three provinces: Inner Mongolia has 5 measuring points, Jilin has 4 measuring points, and Heilongjiang has 3. After threshing, samples were homogenized by a homogenizer and frozen for service. 

### 2.3. Quantification of Protein

A total of 1 g of the sample, 0.4 g of copper sulfate solution, and 6 g of potassium sulfate solution were mixed in a digestion tube and heated to 420 °C, and then continued to heat in the tube for 1 h. When the sample was a green color, it was cooled to room temperature and 50 mL of water was added. Then, sodium hydroxide, hydrochloric acid, and boric acid were mixed with 2 portions of methyl red (1 g/mL) and a portion of methylene blue (1 g/mL); the sample was mixed and transferred to the automatic Kjeldahl apparatus.

The content of the sample protein was calculated by the following formula: $$X=\left(\left(V_{1}–V_{2}\right)\times c\times 0.0140\right)/\left(m\times \left(V_{3}/100\right)\right)\times F\times 100$$
where:

*X* denotes the content of protein in the sample (g/100 g), *V*_1_ denotes the volume of hydrochloric acid to titrate the sample (mL), *V*_2_ denotes the volume of hydrochloric acid to titrate the blank reagent (mL), *c* denotes the concentration of hydrochloric acid (0.05 mol/L), 0.014 means the conversion coefficient of hydrochloric acid and nitrogen (g), m denotes the weight of the sample (1 g), *V*_3_ denotes the volume of digestion solution (mL), *F* denotes the conversion coefficient of nitrogen to protein (6.25), and 100 means the conversion coefficient.

### 2.4. Quantification of Amino Acid Composition

The amino acids in fresh corn were detected according to the method stipulated in national food safety standard GB 5009.124–2016. Two parallel experiments were conducted for each sample.

A total of 2 g of the sample was mixed with 10 mL of hydrochloric acid solution (6 mol/L), then 3 drops of phenol were added and fully mixed. The hydrolysis tube was frozen in the refrigerant for 5 min, the sample was evacuated, and then nitrogen was filled in; the above operation was repeated three times. The sample in the confined tube was hydrolyzed at 110 °C ± 1 °C for 22 h and then cooled to room temperature. The sample solution was filtered into a 50 mL volume flask and water was used to wash the tube to the same flask, diluted with water to volume, and mixed. A 1.0 mL sample was transferred into a 15 mL tube, then the sample was reduced—pressure dried at 40~50 °C. The remains were dissolved by 1 mL of water and reduced—pressure dried. The sample and 1.0 mL of sodium citrate buffer (pH = 2.2) were mixed and filtered by membrane filters (0.22 μm). Equal volumes of samples and mixed amino acid standard liquids were injected and analyzed by the Artemis 6000 amino acids analyzer (Scion, Techcomp, Shanghai, China), equipped with chromatographic column SB–C18 (2.1 m × 100 mm × 1.8 μm. Agilent Technologies Inc., Palo Alto, CA, USA).

The content of amino acids in the standard was calculated by the following formula:$$C_{i}=C_{s}/A_{s}\times A_{i}$$
where:

*C_i_* denotes the content of amino acid *i* of standard(nmol/L), *A_s_* denotes the peak area of amino acid *i* of the standard, *C_s_* denotes the content of amino acid *i* of the sample (nmol/L), and *A_i_* denotes the peak area of amino acid *i* of the standard.

The amino acid content in the sample was calculated by the following formula:$$X_{i}=\left(C_{i}\times F\times V\times M\right)/\left(m\times 10^{9}\right)\times 100$$
where:

*X_i_* denotes the content of amino acid *i* of the sample (g/100 g), *C_i_* denotes the content of amino acid *i* of the standard (nmol/L), *F* means the dilution multiple, and *V* denotes the volume of the standard transferred to the volume flask (mL), *M* denotes the molecular weight of the amino acid *i* (g/mol). Moreover, m denotes the weight of the sample (2 g), and 109 and 100 denote the conversion coefficients.

### 2.5. Amino Acid Nutritional Evaluation

The existing amino acid nutrition estimation can be roughly divided into six categories: amino acid composition analysis, chemical evaluation, flavor evaluation, matching degree, evaluation of amino acid pattern spectrum, and factor analysis. 

#### 2.5.1. Amino Acid Composition Analysis

The amino acid composition analysis classifies amino acids into EAAs and NEAAs, flavor amino acids (FAAs), medicinal amino acids (MAAs), etc. The contents of various amino acids were analyzed and evaluated according to the classification results, and the amino acid composition was compared with that of other plant proteins to analyze the difference between proteins to be tested and other proteins; we then compared their advantages and disadvantages. The diversity of food amino acids refers to the types and contents of EAAs in food. According to the ideal model of amino acids published by FAO/WHO, if the ratio of EAAs to TAAs is about 40%, and the ratio of EAAs to NEAAs is more than 60%, the protein can be considered high—quality [22].

#### 2.5.2. Chemical Evaluation

Chemical evaluation is a commonly used method for evaluating food proteins. This evaluation method compares the content of EAAs per unit weight of protein with the EAAs in the reference protein model to measure the nutritional value of protein in food [23]. The chemical score (*CS*), amino acid score (*AAS*), essential amino acids index (*EAAI*), biological value (*BV*), fuzzy recognition method of amino acids (*FNAAP* or *U(a, u*)), nutrition index (NI), the ratio of branched amino acids to aromatic amino acids (*F*), and predict protein efficacy ratio (Predict PER) are classified into the chemical evaluation method of amino acids in this article.

The amino acid evaluation includes *CS* and *AAS* [24]. The *CS*, which uses the whole egg protein model, was proposed by the National Institute for Nutrition and Health of the Chinese Center for Disease Control and Prevention (Chinese Academy of Preventive Medicine). The nutritional values of food amino acids were evaluated by comparing the proximity of EAAs in the tested proteins to EAAs in the whole egg protein model. The closer *CS* is to 1, the higher the nutritional value of the protein. When *CS* is less than 1, the amino acid is the limiting amino acid. Moreover, the amino acid with the lowest *CS* is called the first limiting amino acid. It is believed that the amino acid pattern in whole egg protein is closest to human needs, and foods with higher *CS* values are more easily absorbed by the human body. *CS* is calculated by the following formula:$$CS=AA_{Fi}/AA_{Ei}$$
where: 

*AA_Fi_* means the amount of a certain EAA in the sample, *AA_Ei_* means the amount of certain EAA in the whole egg protein

*AAS* is the ratio of the amount of an amino acid in a sample to the amount of the same amino acid in the FAO/WHO—recommended ideal protein [25]. *AAS* was proposed by WHO and FAO in 1973 to measure whether the protein in a food is nutritious enough for humans. This amino acid model can improve the true utilization rate of protein, plays a role of *AAS* in nutrition evaluation, and provide a theoretical basis for nutrition recipes, protein complementation, and strengthening of EAAs [26,27]. The closer *AAS* is to 100, the higher the nutritional value of the food is. The amino acid with *AAS* of less than 100 is a limiting amino acid, and the amino acid with the lowest *AAS* value is a first—limiting amino acid. *AAS* is calculated by the following formula:$$AAS=AA_{Fi}/AA_{Xi}\times 100$$
where:

*AA_Fi_* means the amount of a certain EAA in the sample, and *AA_Xi_* means the amount of a certain EAA recommended by the FAO/WHO model protein.

Unlike *CS* and *AAS*, *EAAI* evaluates the overall protein quality of food rather than the quality of a certain amino acid, which is a geometric mean model that measures the overall situation of essential amino acids in a sample compared with that in a model protein [28]. *EAAI* considers the ratio of all essential amino acids in the sample protein to those in standard egg protein [29,30]. *EAAI* represents the degree of similarity between EAAs in food and that in standard proteins. When *EAAI* is between 86 and 95, the food is a good protein source. When the *EAAI* is between 75 and 86, the sample is the available protein source. Moreover, when the *EAAI* is less than 75, the food is an unsuitable protein source. The higher *EAAI* is, the more balanced the amino acid composition is, and the higher the protein quality and utilization are. The amino acid ratio coefficient method can directly reflect the degree of amino acid deviation from the pattern spectrum and its influence on the balance of food amino acids [31].
$$EAAI=\sqrt[{i}]{(AA_{F1}/AA_{E1})\times (AA_{F2}/AA_{E2})\times \ldots \times (AA_{Fi}/AA_{Ei}})\times 100$$
where:

*AA_Fi_* means the amount of a certain EAA in the sample (g/100 g protein), *AA_Ei_* means the amount of a certain EAA in the standard of egg protein (g/100 g protein), and *i* mean the number of amino acid species.

*BV* means the amount of protein in the human body converted per 100 g of food protein [29,30]. *BV* represents the degree of digestion and absorption of protein in the human body. The higher the *BV*, the higher the utilization of protein after digestion and absorption [32]. The *BV* value is usually obtained through animal experiments on the digestion and absorption of protein and it also spends a lot of time. Therefore, some scholars are looking for a simpler and faster method to calculate *BV* [33]. There is a significant correlation between *BV* and *EAAI*, and *BV* can be calculated by *EAAI* according to a specific proportional relationship. *BV* can be calculated by the following formula: BV=1.09×EAAI–11.7

The nutritional index is calculated by *EAAI*; this index evaluates (more comprehensively and scientifically) the nutritional value of protein [30]. The *NI* value represents the essential amino acid index adjusted for the protein percentage. Due to the large difference in the protein content of different foods, the percentage of the protein content in food can be used to correct (more strictly) the evaluation results and effectively avoid the error caused by the difference in protein content. The *NI* is calculated by the following formula: NI=EAAI×P/100
where: 

*P* means the percentage of protein in food.

The fuzzy recognition method of amino acids is used to calculate the degree of closeness between protein in food and the standard protein pattern spectrum by means of the Langerian distance. The amino acids to be evaluated are defined as “*u*”, and the model amino acid of the FAO/WHO whole egg protein is “*a*”. The closeness degree between the amino acid to be calculated and the model amino acid *U(a, u)* is calculated by the Langerian distance method. The value of closeness is between 0 and 1, the higher the value is, the closer the amino acid composition of EAA is to the standard protein. The *U(a, u)* to amino acid refers to the close degree between the ratios of all types of EAAs contained in food and the ratios of all EAAs required by the human body. *U(a, u)* can be calculated by the following formula: $$U\left(a,u\right)=1–c\times \overset{7}{\underset{k=1}{\sum }}\left(\left|a_{k}–u_{k}\right|\right)/\left(a_{k}+u_{k}\right)$$
where:

*a_k_* means the content of a certain amino acid in food, *u_k_* means the content of a certain amino acid in the model protein, and c is usually 0.09; thus, the final calculated ambiguity is between 0 and 1, which is easier to compare [34].

Branched—chain amino acids can stimulate the release of insulin and growth hormone, promote substance anabolism, and they have health effects on the human liver. *F* refers to the ratio of branched amino acids to aromatic amino acids in food [35]. The higher the *F* value is, the higher the potential biological activity of the protein [36]. F is calculated by the following formula: $$F=\left(I+L+V\right)/\left(M+T\right)$$
where:

*I* means the content of Ile in food, *L* means the content of Leu in food, *V* means the content of Val in food, *M* means the content of Met in food, and *T* means the content of Tyr in food.

The protein efficiency ratio (*PER*) is one of the oldest methods used to measure the protein quality of food [37]. *PER* is commonly used to describe the net availability of proteins in animal studies, the experimental process is complex and time—consuming. Therefore, Alsemeyer proposed the concept of predicting *PER*; that is, using the contents of each amino acid and regression equation to calculate the predicted value of *PER*. The predicted *PER* value can be obtained more easily and quickly to evaluate the efficacy of the food protein [38].
$$Predict\;PER\left(1\right)=–0.684+0.456M_{Leu}–0.047M_{Pro}\;Predict\;PER\left(2\right)=–0.468+0.454M_{Leu}–0.105M_{Tyr}\;Predict\;PER\left(3\right)=–1.816+0.435M_{Met}+0.780M_{Leu}+0.211M_{His}–0.944M_{Tyr}$$
where:

*M_i_* means the percentage of a certain amino acid in food protein.

*PDCAAS* can reflect the ability of the food protein’s requirement for EAAs [39,40]. This indicator is a protein quality assessment method that was proposed by the FAO/WHO joint expert assessment group in 1989. The true digestibility of a certain food is used to correct the *AAS* value of the food so that the evaluation result is more accurate and consistent with the actual nutritional status of the human body. *PDCAAS* is the *AAS* value corrected by protein digestibility, which can reflect the nutritional value of food amino acids more scientifically. *PSCAAS* is calculated by the following formula:$$PDCAAS=\left(TD\times AAS\right)/100$$
where: 

*TD* means the true digestibility of food, and *AAS* means the amino acid score of the same food.

#### 2.5.3. Amino Acid Ratio Coefficient Method

The proportion of essential amino acids commonly found in the organism is called the pattern of essential amino acid composition. The amino acid pattern spectrum evaluation is a method used to evaluate and analyze various amino acids in food by comparing the amino acid pattern spectra of various standard proteins, also known as the amino acid ratio coefficient method. The protein evaluation is mainly based on a balanced model spectrum. Currently, the commonly used model spectrum in the field of nutrition is the FAO/WHO model jointly proposed by WHO/FAO/UNU, the IOM model spectrum published by the Institute of Medicine of the National Academy of Sciences of the United States, and the whole egg protein model spectrum commonly used in China [40,41]. One of the main methods to analyze the nutritional characteristics of amino acids is to evaluate the parameters of food amino acids based on the essential amino acid balance pattern spectrum [42]. WHO, FAO, and IOM comprehensively considered protein biological availability and optimized the model according to the population’s protein intake demand and protein absorption in the food matrix, to evaluate the nutritional value of food protein more reasonably. 

The amino acid ratio coefficient method evaluates food protein by calculating the ratio of amino acid(s) (*RAA*), the amino acid ratio coefficient (*RC*), and the amino acid ratio coefficient score (*SRC*) in the sample [25]. RC is used to evaluate the balance of food amino acids; the value can reflect the degree of deviation between the food amino acid composition and the amino acid model [42]. The closer the value of each essential amino acid in food is to 1, the closer the content of the essential amino acid is to human needs. If the value of *RC* is larger than 1, the amino acid in food is relatively excessive. If the value of *RC* is less than 1, the amino acid in food is relatively insufficient [43]. *SRC* represents the relative nutritional value of the protein [44,45] and the discrete degree of *RC*. The higher *SRC* is, the smaller the dispersion of the amino acid composition in food, the more balanced the ratio, and the higher the protein utilization rate. The more concentrated the *RC* value is, the smaller the negative contribution of amino acids to its equilibrium, the lower the coefficient of variation, and the larger the *SRC* value. *RAA*, *RC*, and *SRC* are calculated by the following formulas:$$RAA=AA_{Si}/AA_{Pi}\;RC=RAA/\bar{RAA}\;SRC=100–cv\times 100$$
where:

*AA_Si_* means the content of amino acids in a sample and *AA_Pi_* means the content of amino acids in the amino acid pattern spectrum. $\bar{RAA}$ means the average *RAA*, *CV* means the variable coefficient of *RC*.

The values of *RC* and *SRC* obtained by different reference modes are different [46]. Therefore, the limiting amino acids of the same sample calculated by different amino acid pattern spectra may be different, which is mainly due to the difference in the focus of different amino acid pattern spectra. Thus, food evaluations based on FAO/WHO, IOM, and whole egg protein models can ‘understand’ the restricted amino acids in food from different angles and make a comprehensive evaluation of food. Table 1 shows different amino acid pattern spectra of essential amino acids.

Chemical evaluation indices, such as *AAS*, are mainly used for the evaluation of EAAs, concerning the comprehensiveness and richness of food; the *RC* is used to evaluate the amino acid balance [47]. The results of chemical evaluations and amino acid ratio coefficient evaluations can be more comprehensive to understand and evaluate the nutritional value of food protein and provide the scientific basis for a balanced dietary collocation.

#### 2.5.4. Matching Degree

Matching degree (*DM*) refers to the matching degree between the content of EAAs in food and the average daily requirement of amino acids for the human body [48]. *DM* can be calculated by the following formula:$$DM=1/n\times \overset{n}{\underset{i=1}{\sum }}EAA_{i}/EAR_{i}\times 100$$
where: 

*n* means the number of the kind of EAAs in fresh corn, *EAA_i_* means the content of a certain *EAA* in 100 g of fresh corn, *EAR_i_* means the average daily requirement of a certain *EAA*.

### 2.6. Factor Analysis

Factor analysis is one of the most commonly used methods in the comprehensive nutritional evaluation of agricultural products. Due to the variety of amino acids, it is necessary to reduce the dimensions of various amino acids in the evaluation of amino acids, transform multiple variables into a few comprehensive variables, analyze the main factors from multiple influencing factors, reduce evaluation indicators, and simplify the evaluation process. Principal component analysis (PCA) is a method of factor analysis, which can transform multiple variables into a few unrelated comprehensive indicators under the premise of losing less original data by reducing data dimensions. These components are linear combinations of the original variables, which are unrelated to each other, and can reflect most of the information of the original data [49].

First, the correlation between amino acid composition and content in agricultural products was judged according to the correlation coefficient matrix. After dimensional reduction, PCA was conducted to obtain the characteristic value and variance contribution rate. Components with characteristic values greater than 1 were extracted as the main components to construct a comprehensive evaluation model and calculate the comprehensive score of the amino acid content of fresh corn in the three regions; then, the results were sorted.

### 2.7. Statistical Analysis

Data are presented as means ± standard deviations of two parallel tests. Excel (Microsoft 2017, Redmond, WA, USA) was used for basic data processing and index calculations. All data were subjected to analysis of variance (one–way ANOVA) and were compared using Fisher’s least significant difference (LSD) at a 5% significance level (*p ≤* 0.05). SPSS 21.0 (IBM, Armonk, NY, USA) was used for ANOVA and factor analysis, and the maximum variance method was used as the rotation method. GraphPad Prism 9.0 (GraphPad Software, San Diego, CA, USA) was used to produce the graph.

## 3. Results and Discussion

### 3.1. Amino Acid Profiles of Fresh Corn from the Three Provinces of China

In Table 2 and Table 3, the content of protein in fresh corn 1–1 from Inner Mongolia contained the most protein (40.26 ± 0.35 mg/g), followed by fresh corn 1–2 (40.09 ± 1.79 mg/g), and fresh corn 1–5 (39.14 ± 0.86 mg/g). According to the results of one–way ANOVA, the protein contents of fresh corn 1–1, 1–2, 1–4, and 1–5 in Inner Mongolia were significantly different from those in other regions (*p* < 0.05). Table 2 shows the amino acid compositions in fresh corn from different regions. It can be seen that there is little difference in the amino acid compositions of fresh corn in the three regions. A total of 17 types of amino acids were detected, respectively, from fresh corn 1–1, 1–3, and 1–5 in Inner Mongolia and 2–2 in Jilin, while 16 types of amino acids were detected from fresh corn 1–2, 1–4 in Inner Mongolia and 2–1, 2–3, 2–4 in Jilin and 3–1, 3–2, 3–3 in Heilongjiang. Moreover, the total amino acid content of fresh corn 3–3 in Heilongjiang was the highest (38.23 ± 3.07 mg/g), followed by fresh corn 1–2 (37.64 ± 1.38 mg/g), 1–5 (36.64 ± 0.35 mg/g), 1–1 (36.41 ± 0.93 mg/g) from Inner Mongolia, which were significantly higher than those in other regions (*p* < 0.05). The most abundant amino acid in fresh corn was Glu (4.90 ± 0.05~8, 33 ± 0.98 mg/g), followed by Leu (3.02 ± 0.33~5.56 ± 0.65 mg/g), and Ala (2.27 ± 0.02~4.48 ± 0.01 mg/g). The content of Trp was the lowest, and it was detected only in fresh corn 1–1, 1–3, 1–5 from Inner Mongolia, and 2–2 from Jilin. Fresh corn from Jilin and Heilongjiang were more similar in terms of amino acid composition and content. The content of Gly in fresh corn 3–1 was the lowest (0.84 ± 0.05 mg/g), which was significantly different from other fresh corn (*p* < 0.05). Tyr content had no significant differences in fresh corn from Inner Mongolia, and fresh corn 3–3 had the highest content of Tyr (1.34 ± 0.09 mg/g) among the 12 samples.

Table 4 shows that the highest essential amino acid content sample was fresh corn 3–3 from Heilongjiang (13.99 ± 1.22 mg/g), which was significantly different from those of 1–3, 1–4, 1–5, and all of the fresh corn from Jilin (*p* < 0.05). According to the ideal protein standard proposed by FAO/WHO, fresh corn 3–2 from Heilongjiang was closer to the ideal protein standard and had higher nutritional value than fresh corn from other places. Moreover, the ratio of fresh corn 2–2 was significantly different from the other fresh corn in Jilin (*p* < 0.05). The nutritional value of food protein depends more on the composition of amino acids and how closely the food amino acid profile fits the ideal protein standard pattern. In this result, the E/T of fresh corn was about 35%, while *EAA* /NEAA was about 53%, which is different from commonly recognized amino acid diversity [25]. However, the ratio standard of essential amino acids to total amino acids (E/T) set by FAO cannot apply to protein in plant foods. Thus, the amino acid ratio of high–quality plant proteins is not clear and needs further investigation and analysis. At present, there are no numerous plant proteins that are in accordance with the E/T value, and the same is true for fresh corn.

Amino acids are important nutritional components of maize. Some amino acids have special physiological functions in the human body and are called medicine amino acids. Most of the amino acids in food protein are also flavoring substances. Therefore, evaluating amino acids in fresh corn is a significant part of the nutritional evaluation of fresh corn. Amino acids have a variety of physiological activities and can be used to distinguish the source of substances, so amino acid evaluation is of great significance for the origin tracing and quality evaluation of agricultural products [50,51].

In Figure 1, fresh corn 3–3 from Heilongjiang had the most abundant flavor amino acid content (35.92 ± 3.03 mg/g), followed by fresh corn 1–2 (35.13 ± 1.52 mg/g), 1–5 (33.88 ± 0.29 mg/g), and 1–1 (33.76 ± 1.09 mg/g) from Inner Mongolia. Moreover, the contents of umami amino acids and sweet amino acids in fresh corn 3–3 were less differentiating. Although soluble sugars and phenols are the main sources of corn flavor, the presence of taste amino acids can add layers to the taste of corn. In particular, a small amount of Glu and Asp in corn can give food a delicious taste. Moreover, the flavor amino acid contents of the 12 samples from the highest to the lowest were below: bitter amino acids > sweet amino acids > umami amino acids, which meant that the umami and sweet flavors of fresh corn might primarily come from other flavor compounds, such as phenolic compounds and soluble sugar instead of flavor amino acids. The percentages of umami amino acids in the TAAs of fresh corn were 25.82 ± 0.03~28.32 ± 0.73%, the sweet amino acids in total amino acids were 28.73 ± 0.56~31.97 ± 0.53%, and the bitter amino acids were 33.46 ± 0.40~36.91 ± 0.21%.

As seen in Figure 2, a small number of medicinal amino acids exist in plants, which are of great significance for the organism–life activities and maintaining nitrogen balance. Medicinal amino acids usually include Glu, Asp, Arg, Gly, Phe, Met, Leu, Tyr, and Lys [52]. Total medicinal amino acids in fresh corn were between 16.16 ± 0.20~24.08 ± 2.03 mg/g. Glu is the main excitatory neurotransmitter in the central nervous system, which can regulate the learning function of the brain and improve immune functions [53,54]. Asp can regulate intestinal microflora in the human body; it plays an important role in liver failure and defers the occurrence of tiredness [55]. Arg can promote cell proliferation and blood vessel dilation and, thereby, regulate immunity [56]. Gly reduces inflammation and the risk of infectious disease and can protect heart muscle cells [57,58]. Phe can participate in lipid metabolism and glucose metabolism. Moreover, Met is involved in the composition of hemoglobin and serum and can promote the function of the spleen, pancreas, and lymph, and the breakdown of fat [59]. Leu can promote the synthesis of skeletal muscle protein and enhance human immunity [60]. Tyr can be converted into 5–hydroxytryptamine after oxidation and decarboxylation; it can promote sleep and mental stability [61]. Lys is the most easily deficient in food; it can promote the absorption of calcium and facilitate the formation of collagen and connective tissue [62]. The differences between the 12 samples were analyzed by ANOVA in GraphPad Prism 9.0, and the content of medicine amino acids in fresh corn 3–3 and 2–1, 3–1, 3–2 had extremely significant differences (*p* < 0.0001). The contents of medicine amino acids in fresh corn 1–1 and 2–1, 1–1, and 3–2 were extremely significantly different (*p* < 0.001), and there were also extremely significant differences between the medicine amino acid content in fresh corn 1–2 and 3–1, 1–2 and 3–2, 1–5 and 3–2, and 2–3 and 3–3 (*p* < 0.001). Moreover, the content of medicine amino acids between fresh corn 1–1 and 3–1, 1–2 and 2–3, 1–2 and 2–4, 1–3 and 2–1, 1–3 and 3–1, 1–3 and 3–2, 1–4 and 2–1, 1–4 and 3–1, 1–4 and 3–2, 1–5 and 2–1, and 1–5 and 3–1 were significantly different (*p* < 0.01). There were significant differences between the content of medicine amino acids in fresh corn 1–1 and 2–3, 1–1 and 2–4, 1–3 and 2–4, 1–5 and 2–3, 1–5 and 2–4, 2–1 and 2–2, 2–2 and 3–2, and 2–2 and 3–3 (*p* < 0.05).

### 3.2. Chemical Evaluation

The chemical evaluations of the amino acids included CS, AAS, BV, U(a,u), NI, F, predict PER, and PDCAAS. 

The CS values of EAAs in fresh corn from different regions were obtained by calculating the ratios of dietary amino acids to amino acid patterns of whole egg protein. In Table 5, only the Leu CS value is greater than 1, indicating that the amino acid composition in fresh corn does not completely conform to the whole egg protein pattern. In the whole egg protein model, the first limiting amino acid in fresh corn was Met; the CS values of all amino acids, except Trp, Lys, Met, and Cys were greater than 0.6. When choosing foods, in order to meet the digestion and absorption needs of the human body, it is necessary to eat foods that are rich in sulfur–containing amino acids and Lys to achieve nutritional balance. Moreover, fresh corn 1–3 had the most ideal spectrum among the 12 samples and its CS values were significantly different from fresh corn 3–1 (*p* < 0.05), which had the worst CS values in the 12 samples.

As described in Table 6, the AAS values of Lys, Met, Ile, and Trp were below 100, and the AAS values of the other amino acids were over 110, indicating that the amino acid composition in fresh corn was close to the FAO standard model spectra of amino acids. In addition, according to the FAO/WHO model, the first limiting amino acid in fresh corn was Lys, which was different from the first limiting amino acid calculated by CS. The reason was mainly due to the difference in the reference model. Fresh corn 3–2 had the most ideal spectrum of several amino acids except Lys. The highest AAS value of Lys was fresh corn 2–3, which was significantly different from that of fresh corn 3–2 (*p* < 0.05). Moreover, dietary supplementation should also be accompanied by foods rich in Lys and sulfur, such as dairy products, to supplement an amino acid deficiency.

Table 7 shows that the EAAI values of most fresh corn are greater than 80; in particular, the EAAI value of fresh corn 3–3 from Heilongjiang was the highest, indicating that fresh corn has excellent nutritional protein value and could be used as a high–quality source of protein for human health. Moreover, the BV values of most fresh corn were above 80, indicating that fresh corn has high digestion and absorption efficiencies in the human body and it is easy to be used by the human body. The nutritional value was also relatively rich, which was suitable for the daily consumption of residents to supplement nutrients in the body. Fresh corn 1–1 from Inner Mongolia had the closest U(a,u) to 1, which also meant that the protein of fresh corn from Inner Mongolia was the closest to the model protein. Similarly, the U(a,u) of fresh corn was above 0.75, indicating that fresh corn protein is close to the model protein—it is easy to digest and is absorbed by the human body. Numerically, the higher the NI value is, the better the nutritional value of the food. Table 7 shows that the NI value of fresh corn is between 0.021 ± 0.021 and 0.034 ± 0.021. According to the available literature, the F value in fish is between 2 and 3. The normal value of branched aromatics in the human body is between 3 and 4, while the value of branched aromatics in liver disease patients has decreased. Branched aromatics represent the biological activities of protein in food. According to Table 7 and the reported literature, the branched aromatic value (F) of fresh corn is between 3.38 ± 0.00 and 4.17 ± 0.17, which is higher than that of fish [63], but still within the normal range. Therefore, the biological activities of proteins in fresh corn are good and could protect the liver after consumption. Fresh corn is a good protein source that could be used to supplement various amino acids in the human body.

The higher the PER is, the easier the protein is absorbed and utilized. The predict PER of high–quality protein is usually around 2.00. In Table 8, the three predict PERs of fresh corn 3–3, 1–2, 1–4, and 1–2 were higher than the other samples, indicating that the net utilization rate of protein in fresh corn from Inner Mongolia was relatively high, and it was easy to be absorbed and utilized by the human body, while the predict PER value of fresh corn in Jilin province was lower. A comprehensive evaluation of BV, AAS, the ratio of EAAs to TAAs, and the predict PER should be taken into consideration when analyzing the overall digestion and absorption efficiency of the food protein. As these indicators were calculated and analyzed from different perspectives, the comprehensive analysis could make up for the deviation of some evaluation methods and carry out a more reasonable analysis and evaluation of food protein.

PDCAAS is the amino acid score corrected by protein digestibility. Since the protein digestibility of a specific food is introduced into the calculation, the quality of a certain food can be evaluated, and the results can more accurately reflect the ability of the protein to provide the essential amino acid requirements of the human body. The method of evaluating PDCAAS is similar to CS and AAS. The calculated results are compared with 100 to know which one is bigger. The closer the PDCAAS value is to 100, the more the food conforms to the FAO/WHO standard model. In Table 9, the PDCAAS values of Ile, Lys, Val, Trp, and sulfur–containing amino acids are less than 100 and belong to restricted amino acids. This result is different from the evaluation result of AAS, indicating that the types of restricted amino acids can be determined more rigorously through the correction of the true digestibility of food. For the first limiting amino acid, the results are unchanged despite the correction of true digestibility.

In the chemical evaluation of amino acids, the balance and suitability of the food protein can be evaluated more comprehensively by combining multiple indices. According to the evaluation results of fresh corn protein, there are differences in the types of the first limiting amino acids calculated by CS, AAS, and PDCAAS. The first limiting amino acids calculated by CS were sulfur—containing amino acids, while the first limiting amino acid calculated by AAS and PDCAAS was Lys. However, the values of Lys and Met in the three indices were low, which indicated that the differences between the amino acids in fresh corn and the whole egg protein model and the FAO/WHO model were due to these amino acids. From the principle of the amino acid balance, when eating fresh corn, food rich in Lys and sulfur—containing amino acids should be eaten at the same time to make up for the deficiency of fresh corn. EAAI is usually used to evaluate the value of the food protein. Overall, the net utilization rate of fresh corn in the human body is ideal and can be better used by humans.

### 3.3. Evaluation of Amino Acid Pattern Spectrum

The amino acid pattern spectrum evaluation is an evaluation method for the analysis of food amino acids according to different standard pattern spectra. The amino acid pattern spectrum evaluation includes three indices: RAA, RC, and SRC. At present, there are three commonly used pattern spectra, which are the EAA pattern spectrum provided by FAO/WHO, the EAA pattern spectrum specified by IOM, and the standard egg protein pattern spectrum. The significance of the pattern spectrum evaluation is that food protein can be evaluated comprehensively and systematically by comparing it with different standard protein pattern spectra, to obtain more objective evaluation results.

#### 3.3.1. FAO/WHO Standard Pattern Spectrum

The FAO/WHO standard pattern spectrum is usually the most important indicator for the balanced assessment of food amino acid compositions. It can be seen from Table 10 that the FAO/WHO model is used as the standard model spectrum to evaluate the amino acids in food, and the amino acids with low RAA values in the samples were Lys, Trp, and sulfur—containing amino acids, which were mutually verified with the results of the chemical evaluation method. From the point of view of RC value, Ile, Lys, Thr, Trp, and sulfur—containing amino acid contents of fresh corn were insufficient; in particular, the Lys and Met contents in fresh corn are seriously deficient, which would lead to the poor equilibrium of the amino acid composition of corn, affecting the absorption of amino acids (from corn) in humans. The SRC value of fresh corn was relatively balanced except for fresh corn 3–3, indicating that the dispersion of RC in most fresh corn was low and the protein quality was balanced. The SRC values of fresh corn 3–1 and 3–3 in Heilongjiang were lower, indicating that the amino acid composition of fresh corn in Heilongjiang had a high degree of dispersion, a relatively low degree of fitting with the pattern spectrum and that the nutritional value was worse than the other regions.

#### 3.3.2. IOM Model

The IOM model is a new amino acid balance model spectrum proposed by the Institute of Medicine of the American Academy of Sciences. According to the results in Table 11, only the RAA values of Lys, Trp, and Met in the IOM model were less than 1, indicating that the amino acid pattern of fresh corn was closer to the IOM model. Moreover, the indices of this model spectra were high, indicating that the amino acid composition of fresh corn was balanced according to the IOM model, which could provide relatively comprehensive amino acids for the human body. The SRC values in most fresh corn were balanced but were higher than the FAO/WHO pattern spectrum, indicating that the dispersions of RC values of fresh corn were lower (closer to the IOM model).

#### 3.3.3. Egg Pattern Spectrum

In Table 12, the RAA values of EAAs, except for Leu, were all less than 1, indicating that the amino acid composition of fresh corn was extremely different from that of the egg amino acid pattern spectrum. It also meant that the amino acid composition of fresh corn was less balanced when compared to the egg amino acid pattern spectrum. As for the RC values, except for Ile, Lys, and Trp, other EAAs were close to 1, indicating that the protein balance in fresh corn was good. SRC values of fresh corn were not ideal, especially fresh corn 3–1 and 3–3. Moreover, the SRC values of fresh corn 2–1, 2–2, 2–3, and 2–4 in Jilin were higher than in other regions.

Overall, the dispersion of amino acid RC values of fresh corn are relatively low, and the composition is balanced, meeting the needs of the human body. When the IOM model was used to evaluate the amino acid quality of fresh corn, the RAA values, except for Lys, Met, and Trp, were sufficient, indicating that the amino acid content of fresh corn in the IOM model was relatively sufficient, with rich varieties and balanced distributions. According to the RC and SRC values, the amino acid content and composition distribution of fresh corn are relatively balanced and can meet amino acid needs in daily life. Compared with the FAO/WHO model, the evaluation result of the IOM model is more ideal and closer to the nutritional value of fresh corn itself. The calculation results of three parameters of egg amino acid pattern spectra are not ideal, indicating that there is still a considerable gap between fresh corn and high animal—based protein foods in terms of protein content and quality, which also indicates that plant—based protein foods still cannot replace animal protein at present. According to the evaluation results, Ile and Trp are amino acids with the highest deficiency degrees in the egg amino acid pattern spectrum of fresh corn. In addition, the SRC value evaluated by the egg pattern spectrum was the lowest among the three patterns, indicating that the dispersion of the RC value evaluated by the egg amino acid pattern spectrum is larger and the balance of amino acids is poor. 

### 3.4. Flavor Evaluation

Amino acids are important nutrients and play a significant role in preventing the deterioration of flavor or the appearance of food. The flavor evaluation of food amino acids should begin with two aspects: the correlation of flavor amino acids and the threshold ratio of taste. By analyzing the correlation between different types of flavor amino acids, the correlation between different taste traits can be explored, and the flavor and nutritional value of corn can be better understood.

In Table 13, there is a significant correlation between the three types of flavor amino acids (*p* < 0.01), so it can be stated that there is a significant positive correlation between the multiple flavors of fresh corn, which can also provide the bases for the evaluations of the flavor traits of corn.

The content threshold (RCT) ratio is also called the taste activity value (TAV), which represents the contribution of amino acids to food flavor. When TAV is greater than 1, it indicates that amino acids contribute to taste presentation. The larger the TAV is, the greater the contribution to taste presentation. As we can see in Table 14, the TAV values of umami amino acids are all greater than 1, and the TAV values of Glu are as high as 20, indicating that umami amino acids contribute significantly to the taste of fresh corn. The TAV value of fresh corn 1–2 was above 1, which meant that fresh corn 1–2 had a richer flavor than the other samples.

### 3.5. Evaluation of Amino Acid Matching Degree

The degree of matching (DM) is used to describe the proximity between the EAA content in food and the daily requirement of human EAA, according to the principle mentioned above, based on every 100 g sample of edible parts providing EAAs, and accounting for the reference intake of dietary nutrients for Chinese people, different people at different ages (regarding the essential amino acids), and the ratio of average daily demand calculation sample. According to the report on Nutrition and Chronic Disease of Chinese Residents, the average weights of Chinese male and female residents over the age of 18 are 69.6 and 59 kg, respectively. For 11–12-year-old adolescents, the weight is 45 kg (male) and 42 kg (female) [64].

In Table 15, the DM of amino acids in fresh corn increased gradually with the increase of age, especially in the same age group, the DM of the average daily requirement of EAAs in females to EAAs in fresh corn was significantly higher than that in males, indicating that fresh corn is more suitable for adults and women to supplement the required amino acids. The amino acid DM of fresh corn is mostly between 10 and 16; although the value is lower than in high—protein foods, such as meat, egg, and milk, it is higher than that of citrus [48]. In daily consumption, one needs to pay attention to nutrition collocation for the rational use of amino acids. According to the evaluation results, the DM value of fresh corn 3–3 in Inner Heilongjiang was the highest among the 12 samples, followed by fresh corn 1–2, 1–1, 1–4, 1–5, and 1–3 in Inner Mongolia, indicating that the amino acids in fresh corn in Inner Mongolia had a high—matching degree with the amino acids needed by the human body.

### 3.6. Factor Analysis

At present, factor analysis is one of the most important methods used to evaluate the quality of agricultural products. Factor analysis can simplify many indices into a few factors, which can lower the correlations among indices. In this experiment, 16 amino acids were simplified into 2 principal components by factor analysis, to the comprehensive score of principal components, and to rank the fresh corn according to nutritional value. Regarding the factor analysis of amino acids in fresh corn, it is necessary to conduct a correlation analysis of amino acids first, to explore whether there is a correlation between different amino acids. After determining the correlations between multiple indicators, PCA could be used to simplify multiple indicators.

#### 3.6.1. Correlation Analysis

SPSS 21.0 was used for the Pearson correlation analysis of amino acids in fresh corn; the analysis results are shown in Table 16. As we can see from the table, there are significant correlations among most amino acids. There were significant correlations between Asp and Thr, Ser, Glu, Gly, Ala, Val, Met, Ile, Tyr, Phe, Lys, and Arg (*p* < 0.01), and there were correlations between Asp and Leu and Pro and Trp (*p* < 0.05). Thr and Ser had correlations with all other amino acids, and Glu was significantly correlated with the other amino acids, except for Lys (*p* < 0.01). Gly, Ala, Met, and the other amino acids, except Phe, were correlated (*p* < 0.01); Val significantly correlated with all other amino acids (*p* < 0.01) and was correlated with Phe (*p* < 0.05). Ile and Tyr correlated with all other amino acids (*p* < 0.05). Moreover, there was no correlation between Leu and Lys (*p* > 0.05), Phe and Pro correlated at a 0.05 level. Lys was significantly correlated with Arg and Trp (*p* < 0.01) and correlated with His (*p* < 0.05). Most of the amino acids in fresh corn were significantly correlated, indicating that there was an overlap in the contribution of various amino acid indices to the nutritional value of fresh corn. The 17 amino acid indices could be classified and simplified by PCA, to eliminate the possible influence of the correlation among the indices.

#### 3.6.2. Principal Component Analysis (PCA)

According to the factor analysis results, the accumulative contribution rate of the first three principal components reached 91.026% (Table 17), which could represent most of the information of all amino acid indices and could be used as a comprehensive index for the evaluation of the amino acid nutritional quality of fresh corn. Since the differences between the initial variance rates of the three principal components were large, it was necessary to use the maximum variance contribution method to rotate during the factor analysis, and finally obtain the rotated eigenvalue, variance contribution rate, and cumulative variance contribution rate. 

As can be seen in Table 17, the contribution rate of PC1′s rotational variance was 46.072%, indicating that PC1 could explain nearly half of the amino acid indicators. Table 17 shows the rotation matrix of the amino acid factor analysis; Leu, Pro, Glu, His, Ile, Ser, Met, Val, Tyr, Thr in PC1 had great loads, so they could describe the characteristic data of the first principal component more completely. The value of Thr was around 0.6, but compared with the loads of the two types of amino acids in the second and third principal component values, the load value of Thr in PC1 was higher, so more information on Thr was loaded in the first principal component. 

The amino acids with high loads in the first principal component were generally the amino acids directly involved in the biological regulation of the human body. Leu helps build muscles, improves exercise, and enhances immunity [65,66]. Moreover, Pro helps build collagen and cartilage, keeps muscles and joints flexible, and helps one maintain healthy skin and connective tissue growth [67]. Glu, as an important neurotransmitter, is involved in various physiological and biochemical processes in the human body and can improve the immune functioning of the human body [68,69]. His can promote the absorption of iron, prevent anemia, reduce the acidity of gastric juice, and has a similar effect on allergic diseases, such as asthma [70,71]. Ile boosts the effects of growth hormone and works with Leu and Val to repair muscles and burn visceral fat [72]. Ser plays an important role in fat and fatty acid metabolism and muscle growth and contributes to the production of immune hemocytes and antibodies [73]. Met, as an essential amino acid, cannot be synthesized in the human body and needs to be ingested from food. It can inhibit the increase of blood pressure and has important physiological functions in the human body [74]. Val can repair tissue, regulate blood sugar, provide energy, and prevent muscle weakness [75]. Tyr promotes the normal functioning of the adrenal, pituitary, and thyroid glands, regulates mood, and speeds up the body’s metabolism [76,77]. Thr is important in the body’s immune system [78]. These amino acids participate in various metabolic processes in the human body and are indispensable nutrients for the human body. Therefore, PC1 was named as a functional principal component.

It can be seen from Table 17 that 33.448% of the variance was assigned to principal component group 2, which means that the rotated principal component 2 can explain about 34% of the characteristic data of all amino acids. Lys, Gly, Ala, Asp, Arg, and Trp showed high positive loads in the second principal component, indicating that these amino acids had more characteristic data in PC2. In the second principal component, most of the amino acids with large load values in the human body have regulatory effects on bodily functions, especially in regulating and improving immunity. Lys can regulate the balance of human metabolism, promote appetite, accelerate bone growth, and improve immunity [79]. Gly is an inhibitory neurotransmitter involved in the synthesis of glutathione [58]. Ala helps to improve the body’s energy, assists in glucose metabolism, and prevents kidney stones [80]. Asp, as an excitatory neurotransmitter, plays a regulatory role in various metabolism cycles in vivo and can regulate intestinal flora [55]. Arg plays an important role in regulating human immune functions and improving immunity [81]. Trp can regulate the metabolic process of lung cancer [82]. It could be seen that the six amino acids with relatively large positive loads in PC2 have, more or less, certain effects on regulating the body, so the second principal component was named the regulatory principal component here.

Phe loaded the most information in PC3, which occupied 11.507% of the information on the fresh corn amino acid composition. Moreover, Phe protects the liver and heart muscle, lowers blood pressure, and plays an important role in biosynthesis and metabolism [83]; thus, PC3 was named the protection principal component.

Table 18 shows the rotational component matrix of the factor analysis; Figure 3 presents the rotational load diagram of the amino acid principal component analysis. Table 19 shows the weight score of each amino acid factor analysis in fresh corn. According to the weight score, the contribution degree of each amino acid to the nutritional value of fresh corn could be more clearly understood, which could be used as one of the evaluation bases in the comprehensive evaluation and analysis. The calculation method of the factor analysis index weight is as follows: first, we calculated the ratio of each index of two principal components in the rotating component matrix to the sum of the indices, then we multiplied the ratio of each amino acid index calculated by the rotation variance contribution rate of the corresponding principal component, and we added the weights of the two principal components to obtain the comprehensive weight of each index. The index weights (from high to low) of the 17 types of amino acids were: Thr (0.0676), Val (0.0668), Ile (0.0662), Ser (0.0653), Glu (0.0634), Tyr (0.0631), Arg (0.0624), Gly (0.0618), Trp (0.0616), Ala (0.0568), Pro (0.0566), Leu (0.0562), Met (0.0558), His (0.0556), Asp (0.0553), Lys (0.0463) and Phe (0.0393).

#### 3.6.3. Comprehensive Evaluation

The scores of the three principal components were calculated by using the standardized data of 17 types of amino acids, and the comprehensive amino acid scores of fresh corn in each region were calculated by using the contribution rate of rotation variance as the coefficient. 

There are many ways to calculate the comprehensive score of the factor analysis. Here, the product sum of the standardized index (Zi) and the corresponding load values in each principal component, respectively, and the comprehensive score, were calculated according to the principal component assignment corresponding to the variance contribution rate. The functions of the two factors are expressed as Y1, Y2, and Y3, respectively.
$$\begin{array}{ll} Y_{1}=0.059\times Z_{\mathrm{Asp}} & +0.240\times Z_{\mathrm{Thr}}+0.277\times Z_{\mathrm{Ser}}+0.311\times Z_{\mathrm{Glu}}+0.220\times Z_{\mathrm{Gly}}\\ & +0.144\times Z_{\mathrm{Ala}}+0.263\times Z_{\mathrm{Val}}+0.274\times Z_{\mathrm{Met}}+0.291\times Z_{\mathrm{Ile}}\\ & +0.332\times Z_{\mathrm{Leu}}+0.259\times Z_{\mathrm{Tyr}}+0.115\times Z_{\mathrm{Phe}}+0.036\times Z_{\mathrm{Lys}}\\ & +0.307\times Z_{\mathrm{His}}+0.212\times Z_{\mathrm{Arg}}+0.326\times Z_{\mathrm{Pro}}+0.172\times Z_{\mathrm{Trp}} \end{array}$$
$$\begin{array}{ll} Y_{2}=0.310\times Z_{\mathrm{Asp}} & +0.281\times Z_{\mathrm{Thr}}+0.224\times Z_{\mathrm{Ser}}+0.128\times Z_{\mathrm{Glu}}+0.312\times Z_{\mathrm{Gly}}\\ & +0.310\times Z_{\mathrm{Ala}}+0.258\times Z_{\mathrm{Val}}+0.197\times Z_{\mathrm{Met}}+0.193\times Z_{\mathrm{Ile}}\\ & +0.042\times Z_{\mathrm{Leu}}+0.182\times Z_{\mathrm{Tyr}}+0.007\times Z_{\mathrm{Phe}}+0.407\times Z_{\mathrm{Lys}}\\ & +0.151\times Z_{\mathrm{His}}+0.309\times Z_{\mathrm{Arg}}+0.083\times Z_{\mathrm{Pro}}+0.309\times Z_{\mathrm{Trp}} \end{array}$$
$$\begin{array}{ll} Y_{3}=0.432\times Z_{\mathrm{Asp}} & +0.207\times Z_{\mathrm{Thr}}+0.169\times Z_{\mathrm{Ser}}+0.249\times Z_{\mathrm{Glu}}+0.046\times Z_{\mathrm{Gly}}\\ & +0.184\times Z_{\mathrm{Ala}}+0.166\times Z_{\mathrm{Val}}+0.004\times Z_{\mathrm{Met}}+0.226\times Z_{\mathrm{Ile}}\\ & +0.212\times Z_{\mathrm{Leu}}+0.282\times Z_{\mathrm{Tyr}}+0.605\times Z_{\mathrm{Phe}}+0.029\times Z_{\mathrm{Lys}}\\ & +0.003\times Z_{\mathrm{His}}+0.097\times Z_{15\mathrm{Arg}}+0.140\times Z_{\mathrm{Pro}}–0.214\times Z_{\mathrm{Trp}} \end{array}$$

Then, the comprehensive score Y was calculated according to the contribution rate of the rotation variance. The formula is as follows:$$Y=0.506\times Y_{1}+0.367\times Y_{2}+0.126\times Y_{3}$$

According to the above formula, the comprehensive evaluation score and ranking of the amino acid factor analysis of fresh corn were important bases for the amino acid evaluation of fresh corn. The higher the principal component score, the higher the amino acid content in fresh corn. As shown in Table 20, from the comprehensive score and ranking, fresh corn 1–2 in Inner Mongolia ranked first, indicating that its amino acid nutritional value was high, the distribution was relatively balanced and suitable to eat, and the score of PC2 in fresh corn 1–2 in Inner Mongolia was higher than that of PC1 and PC3.

According to the comprehensive score of the fresh corn factor analysis, fresh corn 1–2 had the highest score, followed by fresh corn 3–3, 1–1, and fresh corn 3–1 had the lowest score. The results showed that the quality of fresh corn in Inner Mongolia was the best, followed by Jilin and Heilongjiang, from the ‘perspective’ of the principal component analysis of amino acids.

In this experiment, amino acid profiles were used to evaluate the composition of amino acids in fresh corn, the results of the chemical evaluations involved the nutritive quality of fresh corn protein and its digestibility in the human body. The amino acid ratio coefficient method was used for judging how close the fresh corn amino acid composition was to the standard amino acid pattern spectrum, and the amino acid matching degree was used to evaluate how well the fresh corn protein was to human body needs. The factor analysis was used to simplify the evaluation indices and comprehensively evaluate the nutritive quality of fresh corn. In summary, amino acid profiles and the amino acid factor analysis focused on food proteins and amino acids themselves, but chemical evaluations and the amino acid ratio coefficient method focused more on the relationship between food proteins and human needs. Thus, evaluating amino acids in a food could be used via a combination of these different evaluation methods. 

## 4. Conclusions

According to the Chinese food pagoda (regarding a balanced diet), each person should eat between 50 and 150 g of whole grains and beans a day. A single stick of fresh corn can meet the daily dietary requirements of whole grains and beans. On 26 April 2022, the “Dietary Guidelines for Chinese residents (2022)” was published by the Chinese Society of Nutrition; article 8 of the residents’ dietary guidelines proposed that Chinese residents consume more fruits and vegetables, milk, whole grains, and soybeans. Fresh corn belongs to the whole grains category because of its rich vitamins and could act as a fruit or vegetable when eaten. Moreover, it meets the current dietary requirements. 

The protein content of fresh corn is lower than that of meat, eggs, and milk, but higher than that of fruits and vegetables; it can be used as one of the daily dietary sources for amino acid supplementation. The content of Glu in fresh corn is the highest, followed by Leu, Ala, and Pro, which account for more than 50% of the total amino acid content in fresh corn. According to the research results, the E/T of fresh corn is about 35%, while EAA/NEAA is about 53%.

Based on the results of this study, the total amount of flavor amino acids and the various flavor amino acid contents in fresh corn from different regions are different; bitter amino acids are the most, sweet amino acids are second, and umami amino acids are the lowest. The results show that the composition of flavor amino acids in fresh corn is the same, and there is no difference in the order of the flavor amino acid content in different regions. The results of this study show that the percentages of medicinal amino acids in the TAAs of fresh corn are different between different regions; in particular, the percentage of medicinal amino acids in fresh corn 3–3 of Heilongjiang province is the highest among the 12 samples. There is an extremely significant difference between the percentage of medicinal amino acids in fresh corn 3–3 and 3–1 and 3–3 and 3–2 (*p* < 0.0001) from Inner Mongolia, which may be due to the difference in the seeding times. 

From the evaluation results, the amino acid composition of fresh corn and whole egg protein is not similar. Except for Leu, all CS values of EAAs are less than 1, indicating that the limiting amino acid in fresh corn is more (compared with whole egg protein standards), but most of the CS values are above 0.6. The results show that the deviation of the amino acid composition of fresh corn to whole egg protein is small, which could meet the requirements of the human body to a certain extent. AAS corresponds to the FAO/WHO standard model of EAAs. The first limiting amino acid evaluated by different essential amino acid models, combined with the results of CS and AAS, is different. The first limiting amino acid evaluated by CS is Met, and the first limiting amino acid evaluated by AAS is Lys. The evaluation results showed that sulfur–containing amino acids and Lys are limiting amino acids in fresh corn, and other EAAs could meet human needs under the FAO/WHO standard model of EAAs.

According to the chemical evaluation results, the protein content of fresh corn is limited, and it is difficult to provide amino acids for the human body as the main source of dietary protein. However, its composition is close to the FAO/WHO standard model, and its digestion, absorption, and utilization efficiency in the human body are high. Moreover, it is a high—quality protein source with good liver protection effects; moreover, it is a whole grain food that can meet the requirements of human amino acids. The digestion and absorption efficiency of fresh corn in the human body is high and can meet the needs of the human body. However, the ratio of amino acids after protein digestibility correction shows that only one limiting amino acid (Ile) is increased in fresh corn, indicating that the amino acid composition of fresh corn could still roughly meet human needs after digestibility correction.

Regarding the spectra based on the evaluations of three different patterns to evaluate the amino acid of fresh corn—there are differences in the results. However, it can be seen that in the three regions, the RAA values of fresh corn 1–1 in different evaluation modes are ideal, and the RC value and its discrete degree are smaller, showing that the amino acid composition of fresh corn 1–1 is mostly balanced, and is closer to the human body’s needs. 

From the analysis results, there are extremely significant correlations between umami and sweet and bitter amino acids in fresh corn, indicating that the relationship between flavor amino acids and fresh corn is complicated. Analysis results show that the flavor amino acids in fresh corn 1–1 have a stronger value than the taste threshold, but other samples show a low TVA value. The results show that sweet amino acids play a limited role in the taste of fresh corn, and most of the sweet taste of fresh corn may come from a large amount of soluble sugar contained in the corn grains. The matching degree of amino acids in fresh corn is different for different people groups, but the matching degree of amino acids in fresh corn is higher than the amino acid requirements of females and adults, indicating that it could meet the amino acid requirements of adult females to a greater extent. To summarize, fresh corn is effective at meeting the matching degree of human essential amino acid requirements.

The results of the principal component analysis simplify the characteristics into three principal components: functional principal component, regulatory principal component, and protection principal component. The calculation of the index weight is also an important part of the factor analysis. By calculating the weight of each index in each principal component, the influence of each amino acid on the nutritional value of fresh corn can be understood, thus providing data support for the balanced collocation of amino acid intake. The ultimate goal of the factor analysis is to make a comprehensive evaluation and rank the research objects in different varieties or regions. The comprehensive evaluation formula of amino acids in fresh corn can be used to calculate the comprehensive score of the amino acid factor analysis of fresh corn. The results show that the amino acid comprehensive score of fresh corn 1–2 is the highest; that is, the amino acid nutritional value is the highest. In conclusion, the protein content of fresh corn 1–2 in Inner Mongolia is the second highest, and the flavor amino acid content is higher than that in other samples, except for fresh corn 3–3. In addition, the EAAI, BV, U(a,u), NI, F, and predict PER evaluation results of fresh corn 3–3 are the highest. Moreover, the factor analysis comprehensive score of fresh corn 1–2 is the best. In conclusion, the amino acid nutritional value of fresh corn 1–2 is the highest. 

## Figures and Tables

**Figure 1 nutrients-14-03887-f001:**
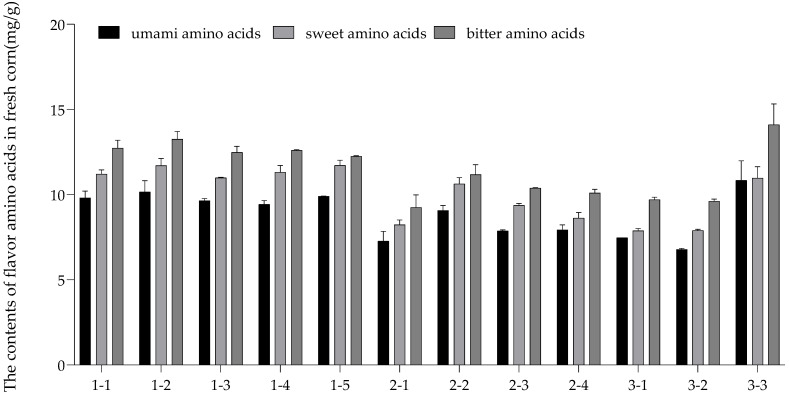
Contents of flavor amino acids in fresh corn of three regions. Glutamic acid (Glu) and aspartic acid (Asp) are umami amino acids; alanine (Ala), glycine (Gly), serine (Ser), proline (Pro), and threonine (Thr) are sweet amino acids; leucine (Leu), isoleucine (Ile), valine (Val), phenylalanine (Phe), methionine (Met), histidine (His), and arginine (Arg) are bitter amino acids.

**Figure 2 nutrients-14-03887-f002:**
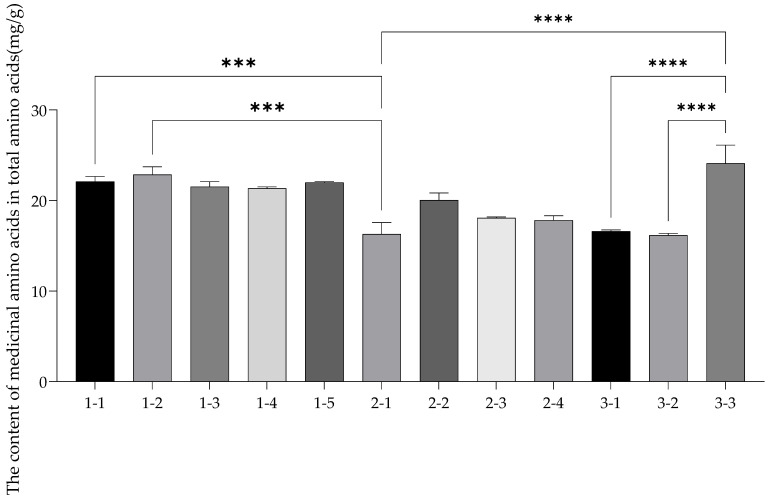
The contents of medicinal amino acids in total amino acids in fresh corn from three regions. The contents of medicine amino acids in fresh corn 3–3 and 2–1, 3–1, and 3–2 had extremely significant differences **** (*p* < 0.0001). Due to limited space, the figure only marks the extreme significant differences **** (*p* < 0.0001) and some significant differences *** (*p* < 0.001) among the medicine amino acid content of the samples.

**Figure 3 nutrients-14-03887-f003:**
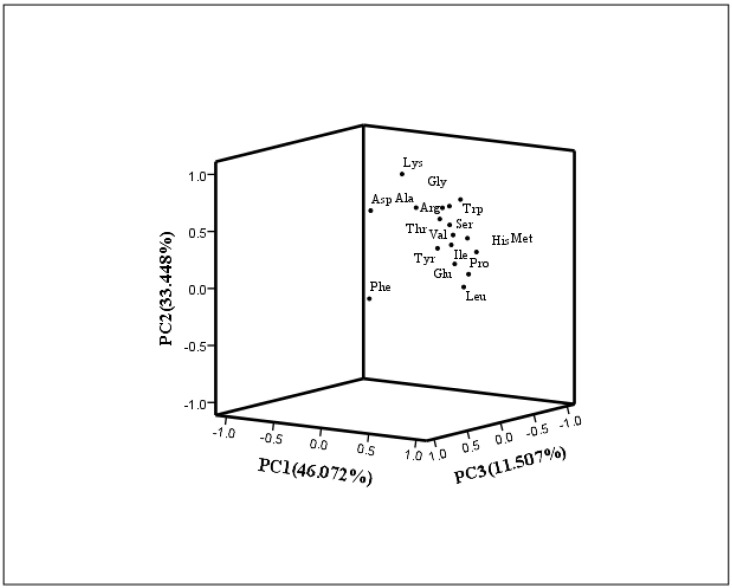
Amino acid principal component analysis rotational load diagram.

**Table 1 nutrients-14-03887-t001:** Three spectra patterns of *EAAs* (mg/g protein).

Pattern Spectrum Amino Acid	His	Thr	Lys	Leu	Ile	Met + Cys	Phe + Tyr	Val	Trp
FAO/WHO/UNU Adults	15	23	45	59	30	22	38	39	6
IOM	18	27	51	55	25	25	47	32	7
Whole Egg	–	47	70	86	54	57	93	66	17

**Table 2 nutrients-14-03887-t002:** Contents of 17 amino acids in fresh sweet–waxy corn from different regions.

Amino Acids	Inner Mongolia					Jilin				Heilongjiang		
	1–1	1–2	1–3	1–4	1–5	2–1	2–2	2–3	2–4	3–1	3–2	3–3
Asp	2.45 ± 0.02 ^abc^	2.33 ± 0.07 ^bcd^	2.24 ± 0.12 ^d^	2.18 ± 0.04 ^d^	2.62 ± 0.02 ^a^	2.25 ± 0.12 ^d^	2.58 ± 0.11 ^a^	2.29 ± 0.05 ^acd^	2.22 ± 0.06 ^d^	1.82 ± 0.02 ^e^	1.88 ± 0.02 ^e^	2.50 ± 0.17 ^b^
Thr	1.37 ± 0.03 ^abc^	1.41 ± 0.01 ^ab^	1.32 ± 0.04 ^bc^	1.31 ± 0.01 ^cd^	1.41 ± 0.02 ^a^	1.01 ± 0.05 ^f^	1.23 ± 0.03 ^de^	1.17 ± 0.02 ^e^	1.15 ± 0.02 ^e^	0.94 ± 0.04 ^f^	1.01 ± 0.01 ^f^	1.37 ± 0.08 ^abc^
Ser	1.82 ± 0.01 ^ab^	1.87 ± 0.09 ^a^	1.79 ± 0.07 ^ab^	1.72 ± 0.06 ^b^	1.83 ± 0.01 ^a^	1.22 ± 0.05 ^e^	1.56 ± 0.01 ^c^	1.46 ± 0.01 ^d^	1.51 ± 0.05 ^cd^	1.30 ± 0.00 ^e^	1.26 ± 0.01 ^e^	1.82 ± 0.05 ^ab^
Glu	7.37 ± 0.42 ^bc^	7.83 ± 0.73 ^ab^	7.40 ± 0.01 ^bc^	7.25 ± 0.18 ^bc^	7.29 ± 0.01 ^bc^	5.03 ± 0.45 ^e^	6.49 ± 0.19 ^cd^	5.58 ± 0.01 ^de^	5.70 ± 0.23 ^de^	5.64 ± 0.02 ^de^	4.90 ± 0.05 ^e^	8.33 ± 0.98 ^a^
Gly	1.36 ± 0.03 ^ab^	1.39 ± 0.02 ^a^	1.27 ± 0.07 ^cde^	1.21 ± 0.02 ^def^	1.31 ± 0.03 ^abc^	1.02 ± 0.04 ^i^	1.18 ± 0.05 ^efg^	1.14 ± 0.02 ^fgh^	1.07 ± 0.05 ^hi^	0.84 ± 0.05 ^j^	1.10 ± 0.01 ^ghi^	1.28 ± 0.01 ^bcd^
Ala	3.57 ± 0.07 ^de^	3.93 ± 0.00 ^bc^	3.64 ± 0.10 ^cd^	4.02 ± 0.04 ^b^	4.48 ± 0.01 ^a^	2.90 ± 0.19 ^g^	3.96 ± 0.18 ^bc^	3.28 ± 0.09 ^ef^	2.60 ± 0.08 ^gh^	2.44 ± 0.06 ^hi^	2.27 ± 0.02 ^i^	3.23 ± 0.36 ^f^
Val	2.02 ± 0.08 ^a^	1.94 ± 0.00 ^a^	1.89 ± 0.09 ^a^	1.86 ± 0.01 ^ab^	1.91 ± 0.02 ^a^	1.41 ± 0.11 ^def^	1.71 ± 0.07 ^bc^	1.57 ± 0.01 ^cd^	1.50 ± 0.05 ^de^	1.29 ± 0.04 ^f^	1.39 ± 0.01 ^ef^	1.92 ± 0.18 ^a^
Met	0.78 ± 0.07 ^a^	0.77 ± 0.02 ^a^	0.71 ± 0.00 ^ab^	0.73 ± 0.04 ^a^	0.72 ± 0.01 ^ab^	0.61 ± 0.07 ^c^	0.58 ± 0.02 ^c^	0.59 ± 0.02 ^c^	0.59 ± 0.01 ^c^	0.47 ± 0.01 ^d^	0.65 ± 0.01 ^bc^	0.78 ± 0.05 ^a^
Ile	1.33 ± 0.05 ^a^	1.32 ± 0.05 ^ab^	1.31 ± 0.09 ^ab^	1.29 ± 0.01 ^ab^	1.32 ± 0.04 ^ab^	0.94 ± 0.10 ^d^	1.16 ± 0.06 ^bc^	1.08 ± 0.02 ^cd^	1.03 ± 0.03 ^cd^	0.95 ± 0.05 ^d^	0.98 ± 0.01 ^d^	1.41 ± 0.16 ^a^
Leu	4.46 ± 0.25 ^bc^	4.92 ± 0.55 ^b^	4.61 ± 0.24 ^b^	4.74 ± 0.04 ^b^	4.33 ± 0.00 ^bc^	3.02 ± 0.33 ^e^	3.86 ± 0.07 ^cd^	3.33 ± 0.02 ^de^	3.25 ± 0.23 ^de^	3.79 ± 0.01 ^cd^	3.27 ± 0.03 ^de^	5.56 ± 0.65 ^a^
Tyr	1.24 ± 0.05 ^a^	1.24 ± 0.00 ^a^	1.33 ± 0.06 ^a^	1.24 ± 0.04 ^a^	1.30 ± 0.01 ^a^	0.93 ± 0.09 ^c^	1.24 ± 0.03 ^a^	1.12 ± 0.02 ^b^	1.08 ± 0.00 ^b^	1.03 ± 0.04 ^bc^	1.02 ± 0.01 ^bc^	1.34 ± 0.09 ^a^
Phe	1.50 ± 0.07 ^bc^	1.47 ± 0.08 ^bcd^	1.28 ± 0.08 ^cd^	1.47 ± 0.05 ^bcd^	1.47 ± 0.06 ^bcd^	1.26 ± 0.03 ^cd^	1.53 ± 0.16 ^b^	1.49 ± 0.15 ^bc^	1.53 ± 0.12 ^b^	1.41 ± 0.02 ^cd^	1.24 ± 0.05 ^d^	1.81 ± 0.14 ^a^
Lys	1.41 ± 0.10 ^ab^	1.27 ± 0.13 ^bc^	1.19 ± 0.11 ^c^	1.11 ± 0.02 ^cd^	1.46 ± 0.05 ^a^	1.00 ± 0.04 ^de^	1.23 ± 0.07 ^c^	1.22 ± 0.05 ^c^	1.12 ± 0.01 ^cd^	0.67 ± 0.08 ^f^	0.95 ± 0.01 ^e^	0.96 ± 0.05 ^de^
His	1.10 ± 0.01 ^ab^	1.18 ± 0.04 ^a^	1.19 ± 0.01 ^a^	1.12 ± 0.01 ^ab^	1.02 ± 0.04 ^bc^	0.84 ± 0.01 ^e^	0.98 ± 0.10 ^cd^	1.01 ± 0.05 ^bc^	0.95 ± 0.02 ^cd^	0.88 ± 0.08 ^de^	0.93 ± 0.01 ^cde^	1.12 ± 0.05 ^ab^
Arg	1.54 ± 0.05 ^ab^	1.65 ± 0.14 ^a^	1.49 ± 0.04 ^abc^	1.39 ± 0.09 ^bcd^	1.49 ± 0.04 ^abc^	1.16 ± 0.10 ^e^	1.36 ± 0.09 ^cd^	1.30 ± 0.01 ^de^	1.26 ± 0.03 ^de^	0.92 ± 0.05 ^f^	1.15 ± 0.01 ^e^	1.50 ± 0.02 ^abc^
Pro	3.10 ± 0.22 ^ab^	3.12 ± 0.35 ^ab^	2.97 ± 0.31 ^ab^	3.07 ± 0.29 ^ab^	2.68 ± 0.23 ^bc^	2.08 ± 0.06 ^d^	2.71 ± 0.12 ^bc^	2.33 ± 0.04 ^cd^	2.30 ± 0.13 ^cd^	2.36 ± 0.03 ^cd^	2.26 ± 0.02 ^cd^	3.27 ± 0.17 ^a^
Trp	0.25 ± 0.01	–	0.24 ± 0.00	–	0.19 ± 0.01	–	0.13 ± 0.02	–	–	–	–	–
Total acid	36.41 ± 0.93 ^a^	37.64 ± 1.38 ^a^	35.63 ± 0.62 ^ab^	35.70 ± 0.55 ^ab^	36.64 ± 0.35 ^a^	26.69 ± 1.72 ^d^	33.35 ± 1.35 ^b^	29.96 ± 0.12 ^c^	28.84 ± 0.85 ^cd^	26.75 ± 0.38 ^d^	26.25 ± 0.29 ^d^	38.23 ± 3.07 ^a^
Protein	40.26 ± 0.35 ^a^	40.09 ± 1.79 ^a^	34.99 ± 0.61 ^b^	38.78 ± 0.86 ^a^	39.14 ± 0.86 ^a^	32.38 ± 0.45 ^c^	35.16 ± 0.26 ^b^	30.49 ± 0.26 ^cd^	30.74 ± 0.24 ^cd^	35.98 ± 1.17 ^b^	29.18 ± 0.83 ^d^	35.41 ± 1.46 ^b^

Notes: Data are the mean values of two determinations ± SD (mg/g). ^a–j^ Means the contents were significantly different among different regions (*p* < 0.05, ANOVA, Duncan); – means not detected.

**Table 3 nutrients-14-03887-t003:** Percentages (%) of 17 amino acids in fresh sweet–waxy corn from different regions.

Amino Acids	Inner Mongolia					Jilin				Heilongjiang		
	1–1	1–2	1–3	1–4	1–5	2–1	2–2	2–3	2–4	3–1	3–2	3–3
Asp	6.73 ± 0.24	6.21 ± 0.42	6.27 ± 0.22	6.11 ± 0.01	7.15 ± 0.02	8.41 ± 0.08	7.74 ± 0.01	7.64 ± 0.13	7.70 ± 0.01	6.82 ± 0.04	7.16 ± 0.01	6.55 ± 0.07
Thr	3.75 ± 0.17	3.74 ± 0.17	3.71 ± 0.04	3.66 ± 0.09	3.85 ± 0.01	3.78 ± 0.04	3.70 ± 0.06	3.90 ± 0.05	3.99 ± 0.03	3.50 ± 0.12	3.83 ± 0.01	3.59 ± 0.07
Ser	4.99 ± 0.10	4.97 ± 0.06	5.03 ± 0.11	4.80 ± 0.08	5.00 ± 0.01	4.58 ± 0.11	4.68 ± 0.22	4.86 ± 0.02	5.22 ± 0.01	4.87 ± 0.08	4.79 ± 0.01	4.77 ± 0.26
Glu	20.22 ± 0.64	20.79 ± 1.17	20.77 ± 0.34	20.31 ± 0.20	19.89 ± 0.21	18.84 ± 0.46	19.46 ± 0.21	18.62 ± 0.04	19.76 ± 0.23	21.1 ± 0.38	18.66 ± 0.03	21.77 ± 0.80
Gly	3.74 ± 0.18	3.68 ± 0.18	3.55 ± 0.12	3.39 ± 0.01	3.59 ± 0.06	3.83 ± 0.09	3.54 ± 0.01	3.79 ± 0.06	3.70 ± 0.06	3.14 ± 0.16	4.20 ± 0.01	3.36 ± 0.25
Ala	9.82 ± 0.07	10.45 ± 0.38	10.22 ± 0.12	11.25 ± 0.06	12.24 ± 0.10	10.86 ± 0.01	11.86 ± 0.06	10.93 ± 0.27	9.02 ± 0.03	9.11 ± 0.08	8.66 ± 0.02	8.45 ± 0.26
Val	5.55 ± 0.07	5.16 ± 0.19	5.30 ± 0.15	5.20 ± 0.10	5.20 ± 0.10	5.30 ± 0.09	5.13 ± 0.02	5.25 ± 0.05	5.20 ± 0.02	4.83 ± 0.07	5.28 ± 0.01	5.02 ± 0.06
Met	2.15 ± 0.13	2.06 ± 0.01	1.99 ± 0.04	2.05 ± 0.08	1.96 ± 0.02	2.27 ± 0.10	1.75 ± 0.02	1.98 ± 0.08	2.05 ± 0.02	1.75 ± 0.03	2.46 ± 0.01	2.05 ± 0.02
Ile	3.66 ± 0.03	3.51 ± 0.00	3.67 ± 0.19	3.60 ± 0.02	3.61 ± 0.07	3.52 ± 0.14	3.49 ± 0.04	3.61 ± 0.09	3.56 ± 0.01	3.56 ± 0.14	3.73 ± 0.01	3.69 ± 0.12
Leu	12.24 ± 0.38	13.07 ± 0.98	12.93 ± 0.44	13.27 ± 0.09	11.81 ± 0.11	11.31 ± 0.52	11.58 ± 0.27	11.1 ± 0.10	11.25 ± 0.47	14.18 ± 0.24	12.45 ± 0.01	14.53 ± 0.54
Tyr	3.40 ± 0.24	3.29 ± 0.12	3.74 ± 0.10	3.49 ± 0.16	3.54 ± 0.00	3.48 ± 0.13	3.71 ± 0.05	3.73 ± 0.08	3.74 ± 0.11	3.85 ± 0.11	3.89 ± 0.01	3.52 ± 0.04
Phe	4.10 ± 0.09	3.91 ± 0.36	3.61 ± 0.29	4.12 ± 0.20	4.02 ± 0.21	4.74 ± 0.19	4.58 ± 0.29	4.98 ± 0.48	5.31 ± 0.57	5.26 ± 0.00	4.74 ± 0.14	4.73 ± 0.01
Lys	3.89 ± 0.38	3.38 ± 0.48	3.34 ± 0.24	3.12 ± 0.01	3.98 ± 0.09	3.75 ± 0.08	3.67 ± 0.07	4.09 ± 0.19	3.89 ± 0.09	2.50 ± 0.25	3.60 ± 0.01	2.52 ± 0.34
His	3.03 ± 0.10	3.14 ± 0.00	3.34 ± 0.09	3.15 ± 0.02	2.77 ± 0.13	3.15 ± 0.24	2.93 ± 0.18	3.38 ± 0.20	3.29 ± 0.18	3.28 ± 0.25	3.55 ± 0.01	2.94 ± 0.10
Arg	4.24 ± 0.25	4.38 ± 0.54	4.19 ± 0.03	3.90 ± 0.32	4.08 ± 0.07	4.36 ± 0.11	4.08 ± 0.12	4.34 ± 0.00	4.37 ± 0.01	3.44 ± 0.23	4.39 ± 0.01	3.95 ± 0.38
Pro	8.50 ± 0.38	8.27 ± 0.62	8.34 ± 1.02	8.58 ± 0.68	7.30 ± 0.57	7.82 ± 0.71	8.11 ± 0.03	7.78 ± 0.17	7.97 ± 0.21	8.82 ± 0.23	8.59 ± 0.01	8.56 ± 0.24
Trp	0.67 ± 0.03	–	0.58 ± 0.06	–	0.58 ± 0.01	–	0.52 ± 0.02	–	–	–	–	–

Notes: Data are the mean values of two determinations ± SD; – means not detected.

**Table 4 nutrients-14-03887-t004:** EAAs, NEAAs, and ideal protein patterns of fresh corn in three regions.

	Inner Mongolia					Jilin				Heilongjiang		
	1–1	1–2	1–3	1–4	1–5	2–1	2–2	2–3	2–4	3–1	3–2	3–3
EAAs (mg/g)	13.12 ± 0.37 ^ab^	13.40 ± 0.40 ^ab^	12.52 ± 0.50 ^bc^	12.74 ± 0.04 ^b^	12.83 ± 0.03 ^b^	9.44 ± 0.73 ^f^	11.50 ± 0.49 ^cd^	10.66 ± 0.05 ^de^	10.35 ± 0.24 ^ef^	9.64 ± 0.21 ^ef^	9.66 ± 0.13 ^ef^	13.99 ± 1.22 ^a^
EAA/TAA	0.360 ± 0.001 ^abc^	0.356 ± 0.003 ^cde^	0.351 ± 0.008 ^def^	0.357 ± 0.004 ^cde^	0.350 ± 0.002 ^ef^	0.353 ± 0.005 ^cde^	0.345 ± 0.001 ^f^	0.356 ± 0.000 ^cde^	0.359 ± 0.002 ^bcd^	0.360 ± 0.003 ^abc^	0.368 ± 0.001 ^a^	0.366 ± 0.003 ^ab^
EAA/NEAA	0.557 ± 0.003 ^abc^	0.546 ± 0.006 ^cde^	0.537 ± 0.018 ^def^	0.549 ± 0.010 ^cde^	0.534 ± 0.006 ^ef^	0.541 ± 0.011 ^cde^	0.521 ± 0.001 ^f^	0.546 ± 0.000 ^cde^	0.554 ± 0.005 ^bcd^	0.560 ± 0.008 ^abc^	0.576 ± 0.003 ^a^	0.573 ± 0.006 ^ab^

Notes: Data are the mean values of two determinations ± SD. ^a–f^ Means significant differences between different regions. (*p* < 0.05, ANOVA, Duncan).

**Table 5 nutrients-14-03887-t005:** CS values of fresh corn from the three provinces.

	1–1	1–2	1–3	1–4	1–5	2–1	2–2	2–3	2–4	3–1	3–2	3–3
Ile	0.61 ± 0.62 ^bc^	0.61 ± 0.65 ^bc^	0.69 ± 0.63 ^ab^	0.61 ± 0.63 ^bc^	0.63 ± 0.60 ^bc^	0.54 ± 0.06 ^cd^	0.61 ± 0.04 ^bc^	0.66 ± 0.01 ^ab^	0.62 ± 0.01 ^bc^	0.49 ± 0.01 ^d^	0.62 ± 0.02 ^bc^	0.74 ± 0.11 ^a^
Leu	1.29 ± 1.41 ^bcd^	1.43 ± 1.42 ^bc^	1.53 ± 1.44 ^b^	1.42 ± 1.36 ^bc^	1.29 ± 1.24 ^bcd^	1.09 ± 0.13 ^d^	1.28 ± 0.03 ^bcd^	1.27 ± 0.00 ^bcd^	1.23 ± 0.08 ^cd^	1.23 ± 0.04 ^c^	1.30 ± 0.05 ^bcd^	1.83 ± 0.29 ^a^
Lys	0.50 ± 0.48 ^bc^	0.45 ± 0.49 ^cde^	0.49 ± 0.43 ^bcd^	0.41 ± 0.48 ^ef^	0.53 ± 0.49 ^ab^	0.44 ± 0.03 ^de^	0.50 ± 0.03 ^bc^	0.57 ± 0.02 ^a^	0.52 ± 0.00 ^b^	0.27 ± 0.02 ^g^	0.46 ± 0.02 ^cde^	0.39 ± 0.00 ^f^
Met	0.34 ± 0.34 ^abc^	0.34 ± 0.34 ^abc^	0.35 ± 0.34 ^ab^	0.33 ± 0.34 ^bc^	0.32 ± 0.34 ^bc^	0.33 ± 0.04 ^bc^	0.29 ± 0.01 ^c^	0.34 ± 0.01 ^abc^	0.34 ± 0.00 ^abc^	0.23 ± 0.00 ^d^	0.39 ± 0.01 ^a^	0.39 ± 0.04 ^a^
Phe + Tyr	0.73 ± 0.73 ^de^	0.73 ± 0.75 ^e^	0.80 ± 0.79 ^cde^	0.75 ± 0.74 ^de^	0.76 ± 0.77 ^de^	0.73 ± 0.05 ^de^	0.85 ± 0.07 ^abcd^	0.92 ± 0.05 ^ab^	0.91 ± 0.05 ^abc^	0.73 ± 0.00 ^de^	0.84 ± 0.05 ^bcde^	0.96 ± 0.11 ^a^
Thr	0.72 ± 0.75 ^def^	0.75 ± 0.77 ^bcde^	0.80 ± 0.75 ^abc^	0.72 ± 0.74 ^f^	0.77 ± 0.73 ^abcde^	0.66 ± 0.04 ^f^	0.75 ± 0.02 ^bce^	0.82 ± 0.02 ^ab^	0.80 ± 0.01 ^abce^	0.55 ± 0.01 ^f^	0.73 ± 0.03 ^cef^	0.83 ± 0.09 ^a^
Val	0.76 ± 0.75 ^ab^	0.73 ± 0.77 ^ab^	0.82 ± 0.76 ^a^	0.72 ± 0.73 ^ab^	0.74 ± 0.73 ^ab^	0.66 ± 0.06 ^b^	0.74 ± 0.04 ^ab^	0.78 ± 0.00 ^a^	0.74 ± 0.02 ^ab^	0.54 ± 0.00 ^c^	0.72 ± 0.03 ^ab^	0.82 ± 0.11 ^a^
Trp	0.37 ± 0.41 ^ac^	0.43 ± 0.40 ^a^	0.36 ± 0.35 ^ac^	0.36 ± 0.34 ^ac^	0.33 ± 0.33 ^cd^	0.32 ± 0.01 ^cd^	0.32 ± 0.01 ^cd^	0.37 ± 0.01 ^a^	0.36 ± 0.02 ^ac^	0.21 ± 0.04 ^e^	0.36 ± 0.02 ^ac^	0.29 ± 0.02 ^d^

Note: ^a–f^ Means significant differences in the CS values among different regions (*p* < 0.05).

**Table 6 nutrients-14-03887-t006:** AAS values of fresh corn from three regions.

	1–1	1–2	1–3	1–4	1–5	2–1	2–2	2–3	2–4	3–1	3–2	3–3
Ile	110.32 ± 4.82 ^bc^	110.08 ± 8.87 ^bc^	124.44 ± 6.45 ^ab^	110.67 ± 3.43 ^bc^	112.75 ± 0.76 ^bc^	96.99 ± 11.32 ^cd^	110.23 ± 6.65 ^bc^	118.24 ± 1.53 ^ab^	111.24 ± 2.20 ^bc^	88.25 ± 1.99 ^d^	111.89 ± 4.16 ^bc^	133.44 ± 20.42 ^a^
Leu	246.19 ± 15.99 ^bcd^	273.92 ± 42.76 ^bc^	292.64 ± 10.03 ^b^	271.58 ± 8.51 ^bc^	245.74 ± 5.35 ^bcd^	207.6 ± 25.64 ^d^	243.92 ± 5.92 ^bcd^	242.48 ± 0.80 ^bcd^	234.56 ± 14.78 ^cd^	234.36 ± 8.25 ^c^	249.12 ± 9.63 ^bcd^	350.21 ± 55.55 ^a^
Lys	77.98 ± 4.95 ^bc^	70.28 ± 4.18 ^cde^	75.59 ± 5.51 ^bcd^	63.90 ± 2.68 ^ef^	82.80 ± 0.78 ^ab^	68.62 ± 3.91 ^de^	77.44 ± 5.13 ^bc^	89.21 ± 2.94 ^a^	81.07 ± 0.17 ^b^	41.23 ± 3.42 ^g^	72.08 ± 2.70 ^cde^	60.23 ± 0.71 ^f^
Met	88.50 ± 8.35 ^abc^	87.91 ± 6.66 ^abc^	91.92 ± 1.69 ^ab^	85.99 ± 6.72 ^bc^	83.45 ± 0.28 ^bc^	85.35 ± 10.41 ^bc^	75.44 ± 2.84	88.33 ± 2.30 ^abc^	87.24 ± 0.99 ^abc^	59.06 ± 0.22 ^d^	100.83 ± 3.75 ^a^	100.87 ± 11.06 ^a^
Phe + Tyr	178.57 ± 2.70 ^de^	177.63 ± 2.75 ^e^	196.96 ± 5.19 ^cde^	184.22 ± 1.95 ^de^	186.33 ± 7.54 ^de^	178.34 ± 12.64 ^de^	207.15 ± 15.93 ^abcd^	225.53 ± 13.17 ^ab^	223.22 ± 11.79 ^abc^	178.26 ± 1.13 ^de^	204.42 ± 11.18 ^bcde^	234.86 ± 27.30 ^a^
Thr	147.53 ± 1.62 ^cef^	152.57 ± 5.68 ^bce^	164.05 ± 1.80 ^ab^	146.5 ± 1.75 ^ef^	156.7 ± 1.50 ^abce^	135.47 ± 9.19 ^f^	152.36 ± 4.63 ^bce^	166.88 ± 4.06 ^ab^	162.64 ± 2.13 ^abc^	113.14 ± 1.71 ^g^	150.04 ± 5.65 ^bcef^	168.83 ± 17.39
Val	128.75 ± 6.07 ^ab^	124.25 ± 5.61 ^ab^	138.34 ± 3.86 ^a^	122.68 ± 2.26 ^ab^	124.94 ± 3.92 ^ab^	112.08 ± 10.57 ^b^	124.8 ± 6.39 ^ab^	132.43 ± 0.37 ^a^	125.06 ± 3.21 ^ab^	92.08 ± 0.37 ^c^	121.76 ± 4.59 ^ab^	139.34 ± 18.45 ^a^
Trp	103.49 ± 4.37 ^bc^	121.51 ± 4.85 ^a^	102.57 ± 7.98 ^bc^	102.08 ± 0.14 ^bc^	92.23 ± 2.94 ^cd^	91.85 ± 2.73 ^cd^	91.74 ± 3.70 ^cd^	106.07 ± 3.23 ^b^	103.26 ± 5.71 ^bc^	59.24 ± 11.42 ^e^	102.03 ± 5.73 ^bc^	81.76 ± 4.37 ^d^
His	182.66 ± 0.06 ^efg^	197.02 ± 16.23 ^bcdef^	226.76 ± 6.11 ^a^	193.39 ± 6.37 ^cdef^	173.17 ± 10.33 ^fg^	172.42 ± 0.50 ^fg^	185.32 ± 20.14 ^defg^	221.59 ± 10.00 ^ab^	205.51 ± 6.97 ^abcde^	162.43 ± 9.28 ^g^	213.26 ± 8.02 ^abc^	211.73 ± 18.73 ^abcd^

Note: ^a–f^ Means significant differences in the AAS values among different regions (*p* < 0.05).

**Table 7 nutrients-14-03887-t007:** EAAI, BV, U(a,u), NI, and F of fresh corn from three regions.

	1–1	1–2	1–3	1–4	1–5	2–1	2–2	2–3	2–4	3–1	3–2	3–3
EAAI	83.23 ± 1.77	84.99 ± 4.75	90.29 ± 2.98	81.34 ± 2.37	83.03 ± 1.45	74.18 ± 6.60	81.55 ± 4.51	90.67 ± 0.61	86.59 ± 1.65	58.4 ± 1.02	85.22 ± 4.1	91.59 ± 10.67
BV	79.02 ± 1.93	80.94 ± 5.18	86.71 ± 3.25	76.96 ± 2.58	78.80 ± 1.58	69.16 ± 7.19	77.19 ± 4.92	87.13 ± 0.67	82.68 ± 1.8	51.95 ± 1.11	81.19 ± 4.47	88.13 ± 11.63
U	0.89 ± 0.00	0.87 ± 0.01	0.86 ± 0.00	0.87 ± 0.00	0.87 ± 0.00	0.88 ± 0.01	0.85 ± 0.01	0.85 ± 0.01	0.86 ± 0.00	0.83 ± 0.01	0.86 ± 0.01	0.81 ± 0.03
NI	0.03 ± 0.03	0.03 ± 0.03	0.03 ± 0.03	0.03 ± 0.03	0.03 ± 0.03	0.025 ± 0.023	0.030 ± 0.028	0.028 ± 0.028	0.027 ± 0.026	0.021 ± 0.021	0.025 ± 0.025	0.034 ± 0.031
F	3.87 ± 0.00	4.07 ± 0.25	3.82 ± 0.10	3.98 ± 0.02	3.75 ± 0.04	3.50 ± 0.01	3.70 ± 0.01	3.50 ± 0.05	3.46 ± 0.16	4.03 ± 0.10	3.38 ± 0.00	4.17 ± 0.17

**Table 8 nutrients-14-03887-t008:** Predict PER of fresh corn from three regions.

	1–1	1–2	1–3	1–4	1–5	2–1	2–2	2–3	2–4	3–1	3–2	3–3
PER 1	1.20 ± 0.10	1.41 ± 0.23	1.28 ± 0.12	1.33 ± 0.01	1.16 ± 0.01	0.6 ± 0.15	0.95 ± 0.02	0.72 ± 0.01	0.69 ± 0.10	0.93 ± 0.00	0.70 ± 0.01	1.70 ± 0.29
PER 2	1.43 ± 0.12	1.64 ± 0.25	1.48 ± 0.10	1.55 ± 0.02	1.36 ± 0.00	0.81 ± 0.14	1.15 ± 0.03	0.92 ± 0.01	0.89 ± 0.10	1.15 ± 0.01	0.91 ± 0.01	1.92 ± 0.29
PER 3	1.07 ± 0.27	1.44 ± 0.45	1.08 ± 0.13	1.26 ± 0.09	0.86 ± 0.01	0.11 ± 0.20	0.49 ± 0.05	0.19 ± 0.02	0.15 ± 0.18	0.56 ± 0.03	0.25 ± 0.02	1.67 ± 0.69

**Table 9 nutrients-14-03887-t009:** PDCAAS of fresh corn from the three regions.

	1–1	1–2	1–3	1–4	1–5	2–1	2–2	2–3	2–4	3–1	3–2	3–3
Ile	93.77 ± 4.10	93.57 ± 7.54	105.78 ± 5.48	94.07 ± 2.92	95.84 ± 0.65	82.44 ± 9.62	93.70 ± 5.65	100.51 ± 1.3	94.56 ± 1.87	75.02 ± 1.69	95.11 ± 3.53	113.42 ± 17.36
Leu	209.26 ± 13.59	232.83 ± 36.34	248.74 ± 8.52	230.84 ± 7.23	208.88 ± 4.54	176.46 ± 21.79	207.33 ± 5.03	206.11 ± 0.68	199.37 ± 12.56	199.21 ± 7.02	211.75 ± 8.19	297.68 ± 47.22
Lys	66.28 ± 4.20	59.74 ± 3.56	64.25 ± 4.68	54.31 ± 2.28	70.38 ± 0.66	58.32 ± 3.32	65.82 ± 4.36	75.83 ± 2.50	68.91 ± 0.15	35.05 ± 2.91	61.27 ± 2.30	51.2 ± 0.60
Met + Cys	75.23 ± 7.10	74.73 ± 5.66	78.13 ± 1.44	73.09 ± 5.71	70.93 ± 0.24	72.54 ± 8.85	64.13 ± 2.42	75.08 ± 1.95	74.15 ± 0.84	50.2 ± 0.19	85.71 ± 3.19	85.74 ± 9.40
Phe + Tyr	151.79 ± 2.30	150.99 ± 2.34	167.42 ± 4.41	156.59 ± 1.66	158.38 ± 6.41	151.59 ± 10.75	176.08 ± 13.54	191.7 ± 11.20	189.74 ± 10.02	151.52 ± 0.96	173.76 ± 9.50	199.63 ± 23.21
Thr	125.40 ± 1.38	129.68 ± 4.83	139.45 ± 1.53	124.53 ± 1.49	133.2 ± 1.27	115.15 ± 7.81	129.51 ± 3.93	141.85 ± 3.45	138.25 ± 1.81	96.17 ± 1.46	127.54 ± 4.80	143.5 ± 14.78
Val	109.44 ± 5.16	105.61 ± 4.76	117.59 ± 3.28	104.28 ± 1.92	106.2 ± 3.33	95.27 ± 8.98	106.08 ± 5.43	112.56 ± 0.31	106.3 ± 2.73	78.27 ± 0.31	103.5 ± 3.90	118.44 ± 15.68
Trp	87.96 ± 3.72	103.28 ± 4.12	87.18 ± 6.78	86.77 ± 0.12	78.39 ± 2.50	78.07 ± 2.32	77.97 ± 3.14	90.16 ± 2.75	87.77 ± 4.85	50.36 ± 9.71	86.73 ± 4.87	69.49 ± 3.72
His	155.26 ± 0.05	167.47 ± 13.79	192.75 ± 5.19	164.38 ± 5.41	147.2 ± 8.78	146.56 ± 0.42	157.52 ± 17.12	188.35 ± 8.50	174.68 ± 5.93	138.07 ± 7.88	181.28 ± 6.81	179.97 ± 15.92

Notes: the true digestibility of fresh corn is 0.85.

**Table 10 nutrients-14-03887-t010:** RAA, RC, and SRC of FAO/WHO amino acid standard pattern spectra from three regions.

		Ile	Leu	Lys	Met	Phe + Tyr	Thr	Val	Trp
1–1	RAA	0.83 ± 0.04	1.58 ± 0.10	0.64 ± 0.04	0.56 ± 0.05	1.13 ± 0.02	0.85 ± 0.01	1.00 ± 0.05	0.62 ± 0.03
	RC	0.92 ± 0.02	1.76 ± 0.07	0.71 ± 0.06	0.62 ± 0.04	1.26 ± 0.01	0.94 ± 0.03	1.11 ± 0.02	0.69 ± 0.05
	SRC	62.35 ± 2.64							
1–2	RAA	0.83 ± 0.07	1.76 ± 0.27	0.58 ± 0.03	0.55 ± 0.04	1.13 ± 0.02	0.88 ± 0.03	0.97 ± 0.04	0.73 ± 0.03
	RC	0.89 ± 0.02	1.89 ± 0.18	0.62 ± 0.08	0.60 ± 0.01	1.22 ± 0.06	0.95 ± 0.03	1.05 ± 0.02	0.79 ± 0.02
	SRC	58.21 ± 5.93							
1–3	RAA	0.93 ± 0.05	1.88 ± 0.06	0.62 ± 0.05	0.58 ± 0.01	1.25 ± 0.03	0.94 ± 0.01	1.08 ± 0.03	0.62 ± 0.05
	RC	0.95 ± 0.02	1.91 ± 0.02	0.63 ± 0.03	0.59 ± 0.03	1.26 ± 0.07	0.96 ± 0.01	1.09 ± 0.00	0.62 ± 0.03
	SRC	55.9 ± 0.51							
1–4	RAA	0.83 ± 0.03	1.75 ± 0.05	0.52 ± 0.02	0.54 ± 0.04	1.17 ± 0.01	0.84 ± 0.01	0.96 ± 0.02	0.61 ± 0.00
	RC	0.92 ± 0.01	1.93 ± 0.02	0.58 ± 0.01	0.60 ± 0.03	1.29 ± 0.04	0.93 ± 0.01	1.06 ± 0.00	0.68 ± 0.01
	SRC	55.00 ± 0.32							
1–5	RAA	0.85 ± 0.01	1.58 ± 0.03	0.68 ± 0.01	0.52 ± 0.00	1.18 ± 0.05	0.90 ± 0.01	0.97 ± 0.03	0.55 ± 0.02
	RC	0.94 ± 0.02	1.75 ± 0.01	0.75 ± 0.02	0.58 ± 0.01	1.30 ± 0.03	1.00 ± 0.01	1.08 ± 0.01	0.61 ± 0.01
	SRC	61.25 ± 0.82							
2–1	RAA	0.73 ± 0.08	1.33 ± 0.16	0.56 ± 0.03	0.54 ± 0.07	1.13 ± 0.08	0.78 ± 0.05	0.87 ± 0.08	0.55 ± 0.02
	RC	0.89 ± 0.03	1.64 ± 0.06	0.69 ± 0.02	0.66 ± 0.03	1.39 ± 0.02	0.96 ± 0.02	1.08 ± 0.01	0.68 ± 0.08
	SRC	64.15 ± 2.18							
2–2	RAA	0.83 ± 0.05	1.57 ± 0.04	0.63 ± 0.04	0.47 ± 0.02	1.31 ± 0.10	0.88 ± 0.03	0.97 ± 0.05	0.55 ± 0.02
	RC	0.92 ± 0.01	1.74 ± 0.04	0.70 ± 0.01	0.53 ± 0.01	1.45 ± 0.04	0.97 ± 0.02	1.08 ± 0.00	0.61 ± 0.00
	SRC	58.01 ± 0.38							
2–3	RAA	0.89 ± 0.01	1.56 ± 0.01	0.73 ± 0.02	0.56 ± 0.01	1.43 ± 0.08	0.96 ± 0.02	1.03 ± 0.00	0.64 ± 0.02
	RC	0.91 ± 0.02	1.60 ± 0.01	0.75 ± 0.03	0.57 ± 0.02	1.47 ± 0.07	0.99 ± 0.01	1.06 ± 0.01	0.65 ± 0.01
	SRC	62.94 ± 1.52							
2–4	RAA	0.83 ± 0.02	1.51 ± 0.10	0.66 ± 0.00	0.55 ± 0.01	1.41 ± 0.07	0.94 ± 0.01	0.98 ± 0.03	0.62 ± 0.03
	RC	0.89 ± 0.00	1.61 ± 0.08	0.71 ± 0.01	0.59 ± 0.00	1.51 ± 0.10	1.00 ± 0.00	1.04 ± 0.01	0.66 ± 0.03
	SRC	61.75 ± 0.37							
3–1	RAA	0.66 ± 0.01	1.51 ± 0.05	0.34 ± 0.03	0.37 ± 0.00	1.13 ± 0.01	0.65 ± 0.01	0.72 ± 0.00	0.36 ± 0.07
	RC	0.92 ± 0.03	2.10 ± 0.04	0.47 ± 0.05	0.52 ± 0.01	1.58 ± 0.01	0.91 ± 0.03	1.00 ± 0.01	0.50 ± 0.09
	SRC	42.24 ± 0.72							
3–2	RAA	0.84 ± 0.03	1.6 ± 0.06	0.59 ± 0.02	0.63 ± 0.02	1.29 ± 0.07	0.86 ± 0.03	0.95 ± 0.04	0.61 ± 0.03
	RC	0.91 ± 0.00	1.74 ± 0.01	0.64 ± 0.00	0.69 ± 0.00	1.40 ± 0.02	0.93 ± 0.00	1.03 ± 0.00	0.66 ± 0.01
	SRC	61.12 ± 0.07							
3–3	RAA	1.00 ± 0.15	2.25 ± 0.36	0.49 ± 0.01	0.63 ± 0.07	1.49 ± 0.17	0.97 ± 0.10	1.09 ± 0.14	0.49 ± 0.03
	RC	0.95 ± 0.03	2.14 ± 0.08	0.47 ± 0.06	0.60 ± 0.01	1.41 ± 0.01	0.92 ± 0.02	1.03 ± 0.01	0.47 ± 0.03
	SRC	44.05 ± 3.64							

**Table 11 nutrients-14-03887-t011:** RAA, RC, and SRC of IOM amino acid standard pattern spectra from three regions.

		Ile	Leu	Lys	Met	Phe + Tyr	Thr	Val	Trp	His
1–1	RAA	1.32 ± 0.06	2.01 ± 0.13	0.69 ± 0.04	0.78 ± 0.07	1.44 ± 0.02	1.26 ± 0.01	1.57 ± 0.07	0.89 ± 0.04	1.52 ± 0.00
	RC	1.04 ± 0.02	1.58 ± 0.07	0.54 ± 0.05	0.61 ± 0.04	1.13 ± 0.01	0.99 ± 0.03	1.23 ± 0.03	0.70 ± 0.05	1.19 ± 0.03
	SRC	66.38 ± 2.16								
1–2	RAA	1.32 ± 0.11	2.24 ± 0.35	0.62 ± 0.04	0.77 ± 0.06	1.44 ± 0.02	1.30 ± 0.05	1.51 ± 0.07	1.04 ± 0.04	1.64 ± 0.14
	RC	1.00 ± 0.01	1.69 ± 0.15	0.47 ± 0.06	0.59 ± 0.01	1.09 ± 0.06	0.98 ± 0.03	1.15 ± 0.02	0.79 ± 0.02	1.24 ± 0.02
	SRC	63.43 ± 4.62								
1–3	RAA	1.49 ± 0.08	2.39 ± 0.08	0.67 ± 0.05	0.81 ± 0.01	1.59 ± 0.04	1.40 ± 0.02	1.69 ± 0.05	0.88 ± 0.07	1.89 ± 0.05
	RC	1.05 ± 0.04	1.68 ± 0.03	0.47 ± 0.03	0.57 ± 0.02	1.12 ± 0.05	0.98 ± 0.01	1.18 ± 0.01	0.62 ± 0.04	1.33 ± 0.06
	SRC	60.56 ± 0.69								
1–4	RAA	1.33 ± 0.04	2.22 ± 0.07	0.56 ± 0.02	0.76 ± 0.06	1.49 ± 0.02	1.25 ± 0.01	1.50 ± 0.03	0.87 ± 0.00	1.61 ± 0.05
	RC	1.03 ± 0.01	1.73 ± 0.01	0.44 ± 0.01	0.59 ± 0.03	1.16 ± 0.04	0.97 ± 0.01	1.16 ± 0.01	0.68 ± 0.02	1.25 ± 0.01
	SRC	60.75 ± 0.22								
1–5	RAA	1.35 ± 0.01	2.01 ± 0.04	0.73 ± 0.01	0.73 ± 0.00	1.51 ± 0.06	1.33 ± 0.01	1.52 ± 0.05	0.79 ± 0.03	1.44 ± 0.09
	RC	1.07 ± 0.03	1.58 ± 0.00	0.58 ± 0.02	0.58 ± 0.01	1.19 ± 0.02	1.05 ± 0.01	1.20 ± 0.01	0.62 ± 0.01	1.14 ± 0.04
	SRC	65.67 ± 0.66								
2–1	RAA	1.16 ± 0.14	1.70 ± 0.21	0.61 ± 0.03	0.75 ± 0.09	1.44 ± 0.10	1.15 ± 0.08	1.37 ± 0.13	0.79 ± 0.02	1.44 ± 0.00
	RC	1.01 ± 0.04	1.47 ± 0.07	0.52 ± 0.01	0.65 ± 0.03	1.25 ± 0.00	1.00 ± 0.01	1.18 ± 0.02	0.68 ± 0.07	1.25 ± 0.09
	SRC	67.74 ± 1.25								
2–2	RAA	1.32 ± 0.08	2.00 ± 0.05	0.68 ± 0.05	0.66 ± 0.03	1.67 ± 0.13	1.30 ± 0.04	1.52 ± 0.08	0.79 ± 0.03	1.54 ± 0.17
	RC	1.04 ± 0.00	1.56 ± 0.05	0.54 ± 0.01	0.52 ± 0.01	1.31 ± 0.03	1.02 ± 0.03	1.19 ± 0.01	0.62 ± 0.01	1.21 ± 0.06
	SRC	63.01 ± 0.07								
2–3	RAA	1.42 ± 0.02	1.98 ± 0.01	0.79 ± 0.03	0.78 ± 0.02	1.82 ± 0.11	1.42 ± 0.03	1.61 ± 0.00	0.91 ± 0.03	1.85 ± 0.08
	RC	1.01 ± 0.02	1.42 ± 0.00	0.56 ± 0.02	0.56 ± 0.02	1.30 ± 0.07	1.02 ± 0.02	1.15 ± 0.00	0.65 ± 0.02	1.32 ± 0.06
	SRC	66.25 ± 0.45								
2–4	RAA	1.33 ± 0.03	1.92 ± 0.12	0.72 ± 0.00	0.77 ± 0.01	1.80 ± 0.10	1.39 ± 0.02	1.52 ± 0.04	0.89 ± 0.05	1.71 ± 0.06
	RC	1.00 ± 0.01	1.43 ± 0.08	0.53 ± 0.01	0.57 ± 0.00	1.35 ± 0.08	1.03 ± 0.00	1.14 ± 0.02	0.66 ± 0.03	1.28 ± 0.05
	SRC	65.91 ± 0.59								
3–1	RAA	1.06 ± 0.02	1.92 ± 0.07	0.36 ± 0.03	0.52 ± 0.00	1.44 ± 0.01	0.96 ± 0.01	1.12 ± 0.00	0.51 ± 0.10	1.35 ± 0.08
	RC	1.03 ± 0.03	1.87 ± 0.06	0.35 ± 0.03	0.51 ± 0.00	1.40 ± 0.00	0.94 ± 0.02	1.09 ± 0.00	0.49 ± 0.09	1.32 ± 0.08
	SRC	50.59 ± 0.01								
3–2	RAA	1.34 ± 0.05	2.04 ± 0.08	0.64 ± 0.02	0.89 ± 0.03	1.65 ± 0.09	1.28 ± 0.05	1.48 ± 0.06	0.87 ± 0.05	1.78 ± 0.07
	RC	1.01 ± 0.00	1.53 ± 0.00	0.48 ± 0.00	0.67 ± 0.00	1.24 ± 0.02	0.96 ± 0.00	1.12 ± 0.00	0.66 ± 0.01	1.34 ± 0.01
	SRC	65.16 ± 0.06								
3–3	RAA	1.60 ± 0.25	2.87 ± 0.45	0.53 ± 0.01	0.89 ± 0.10	1.90 ± 0.22	1.44 ± 0.15	1.70 ± 0.22	0.70 ± 0.04	1.76 ± 0.16
	RC	1.07 ± 0.04	1.92 ± 0.08	0.36 ± 0.05	0.60 ± 0.00	1.28 ± 0.00	0.97 ± 0.01	1.14 ± 0.02	0.47 ± 0.03	1.19 ± 0.04
	SRC	52.04 ± 3.11								

**Table 12 nutrients-14-03887-t012:** RAA, RC, and SRC of egg amino acid standard pattern spectra from three regions.

		Ile	Leu	Lys	Phe + Tyr	Thr	Val	Trp
1–1	RAA	0.50 ± 0.02	1.52 ± 0.10	0.64 ± 0.04	0.68 ± 0.01	0.67 ± 0.01	0.78 ± 0.04	0.39 ± 0.02
	RC	0.68 ± 0.02	2.05 ± 0.09	0.86 ± 0.07	0.92 ± 0.00	0.90 ± 0.03	1.06 ± 0.03	0.53 ± 0.03
	SRC	50.36 ± 4.10						
1–2	RAA	0.50 ± 0.04	1.69 ± 0.26	0.58 ± 0.03	0.68 ± 0.01	0.69 ± 0.03	0.76 ± 0.03	0.46 ± 0.02
	RC	0.66 ± 0.01	2.21 ± 0.20	0.76 ± 0.10	0.89 ± 0.05	0.90 ± 0.03	0.99 ± 0.02	0.60 ± 0.02
	SRC	44.87 ± 8.26						
1–3	RAA	0.57 ± 0.03	1.80 ± 0.06	0.62 ± 0.05	0.75 ± 0.02	0.74 ± 0.01	0.84 ± 0.02	0.38 ± 0.03
	RC	0.69 ± 0.01	2.21 ± 0.01	0.76 ± 0.03	0.92 ± 0.05	0.91 ± 0.02	1.03 ± 0.00	0.47 ± 0.02
	SRC	43.39 ± 0.28						
1–4	RAA	0.50 ± 0.02	1.67 ± 0.05	0.52 ± 0.02	0.70 ± 0.01	0.66 ± 0.01	0.75 ± 0.01	0.38 ± 0.00
	RC	0.68 ± 0.01	2.26 ± 0.03	0.70 ± 0.02	0.94 ± 0.03	0.89 ± 0.01	1.01 ± 0.00	0.52 ± 0.01
	SRC	41.96 ± 0.92						
1–5	RAA	0.51 ± 0.00	1.51 ± 0.03	0.34 ± 0.48	0.71 ± 0.03	0.71 ± 0.01	0.76 ± 0.02	0.35 ± 0.01
	RC	0.74 ± 0.05	2.18 ± 0.22	0.46 ± 0.65	1.02 ± 0.12	1.02 ± 0.09	1.09 ± 0.12	0.50 ± 0.05
	SRC	39.62 ± 16.03						
2–1	RAA	0.44 ± 0.05	1.28 ± 0.16	0.56 ± 0.03	0.68 ± 0.05	0.61 ± 0.04	0.68 ± 0.06	0.34 ± 0.01
	RC	0.67 ± 0.02	1.94 ± 0.08	0.86 ± 0.02	1.03 ± 0.01	0.93 ± 0.01	1.04 ± 0.01	0.53 ± 0.06
	SRC	54.26 ± 3.60						
2–2	RAA	0.50 ± 0.03	1.50 ± 0.04	0.63 ± 0.04	0.79 ± 0.06	0.69 ± 0.02	0.76 ± 0.04	0.34 ± 0.01
	RC	0.67 ± 0.01	2.02 ± 0.05	0.85 ± 0.02	1.06 ± 0.03	0.92 ± 0.01	1.02 ± 0.00	0.46 ± 0.00
	SRC	50.54 ± 1.58						
2–3	RAA	0.54 ± 0.01	1.49 ± 0.00	0.73 ± 0.02	0.86 ± 0.05	0.75 ± 0.02	0.81 ± 0.00	0.40 ± 0.01
	RC	0.67 ± 0.02	1.88 ± 0.01	0.92 ± 0.04	1.08 ± 0.05	0.94 ± 0.01	1.01 ± 0.01	0.50 ± 0.01
	SRC	56.35 ± 0.19						
2–4	RAA	0.51 ± 0.01	1.45 ± 0.09	0.66 ± 0.00	0.85 ± 0.04	0.73 ± 0.01	0.76 ± 0.02	0.39 ± 0.02
	RC	0.66 ± 0.00	1.89 ± 0.08	0.87 ± 0.02	1.11 ± 0.08	0.96 ± 0.01	1.00 ± 0.01	0.51 ± 0.02
	SRC	55.45 ± 2.17						
3–1	RAA	0.40 ± 0.01	1.44 ± 0.05	0.34 ± 0.03	0.68 ± 0.00	0.51 ± 0.01	0.56 ± 0.00	0.22 ± 0.04
	RC	0.68 ± 0.02	2.43 ± 0.05	0.57 ± 0.05	1.14 ± 0.01	0.86 ± 0.02	0.95 ± 0.01	0.37 ± 0.07
	SRC	31.85 ± 1.68						
3–2	RAA	0.51 ± 0.02	1.54 ± 0.06	0.59 ± 0.02	0.78 ± 0.04	0.68 ± 0.03	0.74 ± 0.03	0.38 ± 0.02
	RC	0.68 ± 0.00	2.06 ± 0.01	0.79 ± 0.00	1.04 ± 0.01	0.91 ± 0.00	1.00 ± 0.00	0.51 ± 0.01
	SRC	49.68 ± 0.26						
3–3	RAA	0.61 ± 0.09	2.16 ± 0.34	0.49 ± 0.01	0.89 ± 0.10	0.76 ± 0.08	0.85 ± 0.11	0.31 ± 0.02
	RC	0.70 ± 0.02	2.48 ± 0.09	0.57 ± 0.08	1.03 ± 0.01	0.88 ± 0.02	0.98 ± 0.01	0.36 ± 0.02
	SRC	30.33 ± 4.30						

**Table 13 nutrients-14-03887-t013:** Correlation analysis of flavor amino acids.

	Umami Amino Acids	Sweet Amino Acids	Bitter Amino Acids
Umami amino acids	1		
Sweet amino acids	0.931 **	1	
Bitter amino acids	0.973 **	0.904 **	1

** Means a significant correlation at the 0.01 level (ANOVA, Duncan).

**Table 14 nutrients-14-03887-t014:** Taste threshold ratios of flavor amino acids.

Amino Acids	Taste Threshold (mg/g)	TVA											
		1–1	1–2	1–3	1–4	1–5	2–1	2–2	2–3	2–4	3–1	3–2	3–3
Umami amino acids													
Asp	1.0	2.45 ± 0.02	2.33 ± 0.07	2.24 ± 0.12	2.18 ± 0.04	2.62 ± 0.02	2.25 ± 0.12	2.58 ± 0.11	2.29 ± 0.05	2.22 ± 0.06	1.82 ± 0.02	1.88 ± 0.02	2.50 ± 0.17
Glu	0.3	24.55 ± 1.4	26.11 ± 2.43	24.67 ± 0.03	24.17 ± 0.62	24.29 ± 0.02	16.77 ± 1.49	21.62 ± 0.64	18.6 ± 0.03	19.01 ± 0.78	18.82 ± 0.07	16.33 ± 0.15	27.78 ± 3.25
Sweet amino acids													
Gly	1.3	1.05 ± 0.02	1.07 ± 0.01	0.97 ± 0.05	0.93 ± 0.02	1.01 ± 0.03	0.79 ± 0.03	0.91 ± 0.04	0.87 ± 0.02	0.82 ± 0.04	0.65 ± 0.04	0.85 ± 0.01	0.99 ± 0.01
Ala	0.6	5.96 ± 0.11	6.55 ± 0.00	6.07 ± 0.17	6.70 ± 0.07	7.47 ± 0.01	4.83 ± 0.31	6.60 ± 0.30	5.46 ± 0.16	4.33 ± 0.14	4.06 ± 0.09	3.79 ± 0.03	5.39 ± 0.60
Ser	1.5	1.21 ± 0.01	1.25 ± 0.06	1.19 ± 0.05	1.14 ± 0.04	1.22 ± 0.01	0.81 ± 0.03	1.04 ± 0.01	0.97 ± 0.01	1.00 ± 0.03	0.87 ± 0.00	0.84 ± 0.01	1.21 ± 0.03
Pro	3.0	1.03 ± 0.07	1.04 ± 0.12	0.99 ± 0.10	1.02 ± 0.10	0.89 ± 0.08	0.69 ± 0.02	0.90 ± 0.04	0.78 ± 0.01	0.77 ± 0.04	0.79 ± 0.01	0.75 ± 0.01	1.09 ± 0.06
Thr	2.6	1.60 ± 1.52	1.61 ± 1.54	1.55 ± 1.48	1.53 ± 1.45	1.53 ± 1.38	1.37 ± 1.35	1.40 ± 1.21	1.41 ± 1.27	1.37 ± 1.22	1.26 ± 1.17	1.30 ± 1.16	1.31 ± 0.90
Met	0.3	2.61 ± 0.22	2.58 ± 0.08	2.36 ± 0.00	2.44 ± 0.14	2.40 ± 0.04	2.03 ± 0.22	1.95 ± 0.06	1.98 ± 0.07	1.97 ± 0.04	1.56 ± 0.04	2.16 ± 0.02	2.61 ± 0.18
Bitter amino acids													
Ile	0.9	1.48 ± 0.05	1.47 ± 0.05	1.45 ± 0.10	1.43 ± 0.01	1.47 ± 0.04	1.05 ± 0.11	1.29 ± 0.07	1.20 ± 0.03	1.14 ± 0.03	1.06 ± 0.06	1.09 ± 0.01	1.57 ± 0.18
Arg	0.5	3.08 ± 0.1	3.29 ± 0.28	2.98 ± 0.08	2.79 ± 0.19	2.99 ± 0.08	2.33 ± 0.21	2.72 ± 0.19	2.60 ± 0.01	2.52 ± 0.07	1.84 ± 0.10	2.31 ± 0.02	3.01 ± 0.05
Leu	1.9	2.35 ± 0.13	2.59 ± 0.29	2.43 ± 0.13	2.49 ± 0.02	2.28 ± 0.00	1.59 ± 0.17	2.03 ± 0.03	1.75 ± 0.01	1.71 ± 0.12	2.00 ± 0.01	1.72 ± 0.02	2.93 ± 0.34
Val	0.4	5.05 ± 0.19	4.85 ± 0.00	4.72 ± 0.21	4.64 ± 0.02	4.77 ± 0.04	3.54 ± 0.28	4.28 ± 0.19	3.94 ± 0.02	3.75 ± 0.13	3.23 ± 0.09	3.46 ± 0.03	4.80 ± 0.44
Lys	0.5	2.83 ± 0.20	2.54 ± 0.26	2.38 ± 0.21	2.23 ± 0.04	2.92 ± 0.09	2.00 ± 0.09	2.45 ± 0.14	2.45 ± 0.10	2.24 ± 0.01	1.34 ± 0.15	1.89 ± 0.02	1.92 ± 0.10
His	0.2	5.52 ± 0.05	5.91 ± 0.22	5.95 ± 0.06	5.62 ± 0.06	5.08 ± 0.19	4.19 ± 0.05	4.89 ± 0.49	5.07 ± 0.27	4.74 ± 0.12	4.39 ± 0.39	4.67 ± 0.04	5.61 ± 0.27
Phe	0.9	1.66 ± 0.08	1.63 ± 0.09	1.43 ± 0.09	1.63 ± 0.06	1.64 ± 0.07	1.41 ± 0.04	1.70 ± 0.18	1.66 ± 0.17	1.70 ± 0.13	1.56 ± 0.02	1.38 ± 0.06	2.01 ± 0.16

**Table 15 nutrients-14-03887-t015:** The degree of matching (DM) in fresh corn from the three regions.

	11–12-Year-Olds		Adults	
	Male	Female	Male	Female
1–1	12.68 ± 0.18	13.59 ± 0.20	11.15 ± 0.15	13.15 ± 0.18
1–2	13.09 ± 0.22	14.02 ± 0.24	11.42 ± 0.17	13.47 ± 0.20
1–3	12.06 ± 0.50	12.92 ± 0.53	10.59 ± 0.43	12.49 ± 0.51
1–4	12.31 ± 0.05	13.19 ± 0.06	10.74 ± 0.01	12.68 ± 0.02
1–5	12.30 ± 0.01	13.18 ± 0.01	10.88 ± 0.04	12.84 ± 0.05
2–1	9.25 ± 0.59	9.91 ± 0.63	8.12 ± 0.54	9.58 ± 0.64
2–2	11.16 ± 0.48	11.95 ± 0.51	9.77 ± 0.40	11.52 ± 0.47
2–3	10.46 ± 0.09	11.21 ± 0.10	9.18 ± 0.04	10.83 ± 0.05
2–4	10.20 ± 0.20	10.93 ± 0.21	8.93 ± 0.18	10.54 ± 0.21
3–1	9.14 ± 0.09	9.79 ± 0.09	7.90 ± 0.14	9.32 ± 0.16
3–2	9.49 ± 0.15	10.17 ± 0.16	8.32 ± 0.12	9.81 ± 0.14
3–3	13.00 ± 1.04	13.93 ± 1.11	11.36 ± 0.88	13.4 ± 1.04

**Table 16 nutrients-14-03887-t016:** The correlation matrix between different amino acids.

	Asp	Thr	Ser	Glu	Gly	Ala	Val	Met	Ile	Leu	Tyr	Phe	Lys	His	Arg	Pro	Trp
Asp	1																
Thr	0.767 **	1															
Ser	0.648 **	0.964 **	1														
Glu	0.595 **	0.892 **	0.940 **	1													
Gly	0.702 **	0.931 **	0.871 **	0.764 **	1												
Ala	0.760 **	0.833 **	0.769 **	0.700 **	0.737 **	1											
Val	0.725 **	0.969 **	0.949 **	0.914 **	0.921 **	0.813 **	1										
Met	0.516 **	0.823 **	0.798 **	0.793 **	0.880 **	0.566 **	0.871 **	1									
Ile	0.675 **	0.947 **	0.947 **	0.959 **	0.866 **	0.756 **	0.968 **	0.857 **	1								
Leu	0.424 *	0.778 **	0.848 **	0.963 **	0.660 **	0.565 **	0.819 **	0.754 **	0.909 **	1							
Tyr	0.656 **	0.903 **	0.901 **	0.893 **	0.780 **	0.748 **	0.890 **	0.675 **	0.935 **	0.837 **	1						
Phe	0.530 **	0.478 *	0.441 *	0.543 **	0.296	0.263	0.429 *	0.301	0.497 *	0.508 *	0.489 *	1					
Lys	0.735 **	0.736 **	0.625 **	0.388	0.790 **	0.739 **	0.687 **	0.523 **	0.548 **	0.181	0.530 **	0.094	1				
His	0.377	0.825 **	0.849 **	0.831 **	0.793 **	0.596 **	0.845 **	0.768 **	0.853 **	0.794 **	0.821 **	0.331	0.465 *	1			
Arg	0.723 **	0.936 **	0.873 **	0.773 **	0.954 **	0.740 **	0.912 **	0.823 **	0.847 **	0.656 **	0.794 **	0.371	0.767 **	0.787 **	1		
Pro	0.434 *	0.787 **	0.858 **	0.927 **	0.716 **	0.591 **	0.839 **	0.781 **	0.875 **	0.912 **	0.787 **	0.460*	0.300	0.837 **	0.694 **	1	
Trp	0.424 *	0.731 **	0.696 **	0.544 **	0.812 **	0.656 **	0.728 **	0.689 **	0.599 **	0.434 *	0.476 *	0.022	0.727 **	0.672 **	0.810 **	0.536 **	1

** Means a significant correlation at the 0.01 level. * Means a significant correlation at the 0.05 level.

**Table 17 nutrients-14-03887-t017:** Principal component eigenvalue and contribution rate.

	Eigenvalue	Variance Contribution Rate (%)	Cumulative Variance Contribution Rate (%)	Eigenvalue of Rotation	Rotation Variance Contribution Rate (%)	The Cumulative Variance Contribution after Rotation (%)
1	12.597	74.103	74.103	7.832	46.072	46.072
2	1.744	10.261	84.363	5.686	33.448	79.519
3	1.133	6.663	91.026	1.956	11.507	91.026

Notes: Maximum variance method was used as the rotation method.

**Table 18 nutrients-14-03887-t018:** Rotational component matrix of the factor analysis.

Amino Acids	Principal Component		
	PC1	PC2	PC3
Asp	0.166	0.739	0.604
Thr	0.673	0.670	0.289
Ser	0.776	0.534	0.236
Glu	0.871	0.305	0.348
Gly	0.617	0.745	0.064
Ala	0.402	0.739	0.258
Val	0.737	0.616	0.232
Met	0.766	0.470	0.005
Ile	0.813	0.461	0.316
Leu	0.930	0.101	0.296
Tyr	0.724	0.434	0.395
Phe	0.321	0.017	0.846
Lys	0.101	0.971	0.041
His	0.860	0.359	0.004
Arg	0.594	0.738	0.136
Pro	0.912	0.197	0.196
Trp	0.480	0.738	–0.299

**Table 19 nutrients-14-03887-t019:** Amino acid factor analysis weight score.

Amino Acids	Weighted Score
Asp	0.0553
Thr	0.0676
Ser	0.0653
Glu	0.0634
Gly	0.0618
Ala	0.0568
Val	0.0668
Met	0.0558
Ile	0.0662
Leu	0.0562
Tyr	0.0631
Phe	0.0393
Lys	0.0463
His	0.0556
Arg	0.0624
Pro	0.0566
Trp	0.0616

**Table 20 nutrients-14-03887-t020:** Comprehensive score and ranking of the amino acid evaluations of fresh corn in three regions.

Regions	Y_1_	Y_2_	Y_3_	Composite Score	Sort
1–1	0.588	0.952	–0.241	0.620	3
1–2	1.186	0.804	–0.919	0.780	1
1–3	1.063	0.144	–0.880	0.480	5
1–4	0.984	–0.092	–0.428	0.410	6
1–5	–0.306	1.459	0.773	0.480	4
2–1	–1.436	–0.082	–0.561	–0.830	11
2–2	–0.791	0.600	1.199	–0.030	7
2–3	–0.954	0.355	0.229	–0.325	8
2–4	–0.909	–0.149	0.293	–0.480	9
3–1	–0.393	–2.251	0.132	–1.010	12
3–2	–0.468	–0.795	–1.507	–0.720	10
3–3	1.435	–0.944	1.909	0.620	2

## Data Availability

The data presented in this study are available upon request from the corresponding author.

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
