# Peer review of "Amino Acid Profiles and Nutritional Evaluation of Fresh Sweet–Waxy Corn from Three Different Regions of China"

_nutrients, 2022, doi:10.3390/nu14193887_

Round 1

Reviewer 1 Report

Authors studied the amino acids compositions and nutritional aspects of Chinese Huangnuo fresh sweet-waxy corn from different regions. In general, the work is interest and the aim is clear. The experimental design and the methods applied are appropriate. The results are well-presented and supported by the discussion. Even though the manuscript is well-written, a moderate English editing is recommended. I have only some minor comments for further manuscript’s improvement.

-L89. Please add ref(s).

-L131. Please revise.

-L391-397. Which rotation was used for PCA?

-L845-947. Conclusions should condensed. Many of those parts should be transferred to the discussion.

Author Response

We would like to thank you so much for your generous compliments and the valuable comments to the manuscript which will help us to improve the quality of this article. We have carefully read and studied your comments, you will find our response to the comments point by point in the file. The comments were also adopted and the relevant changes were made in the manuscript. All made changes in manuscript are marked in red at their respective places.

Notes: All line numbers used in the response letter refer to the latest submitted manuscript.

Point 1: -L89. Please add ref(s).

Response 1: Thank you for your comment. We have added the relevant references in the right place.

Point 2: -L131. Please revise.

Response 2: Thank you for your careful guidance. We have revised relevant content in right place.

Point 3: -L391-397. Which rotation was used for PCA?

Response 3: Thank you very much for your comment. Maximum variance method was used as the rotation method, and we have added relevant notes in right place.

Point 4: -L845-947. Conclusions should condensed. Many of those parts should be transferred to the discussion.

Response 4: Thank you very much for your comment and carefully guidance. We have adapts the conclusions part and moved some content to the discussion part.

Reviewer 2 Report

The comparative analysis of the amino acids compositions of Chinese Huangnuo 9 fresh sweet-waxy corn from three different provinces of China, is clearly discribed, resulting an article with relevant contribution to the research feld. Its a result of hard work and  I congratulate you for that. Even so, I have some reccomandation regarding the article.

The manuscript is clear, with some exceptions, but is relevant for the field and it is presented in a well-structured manner.

The cited references mostly are recent publications and relevant for this paperwork. Does not include excessive number of self-citations.

The manuscript is enough scientifically sound and is the experimental design is appropriate to test the hypothesis of its article. 

The manuscript’s results are reproducible based on the details given in the methods section.

The figures are appropriate, but the table are extremely extended and should be synthetized in a proper manner, because they are not easy to interpret and understand. The data are interpreted appropriately and consistently throughout the manuscript, with the mentions from above. The conclusions are consistent with the evidence and arguments presented.

The ethics statements and data availability statements are enough adequate.

The nutritive evaluation system based on amino acid profiles to evaluate, compare, and rank the fresh sweet-waxy corn planted in different regions, is not enough clearly described. Why you, the authors, use it? Whinch was the reason due to you choose it? So, it needs more explanations from you.

Author Response

We would like to thank you so much for your generous compliments and the valuable comments to the manuscript which will help us to improve the quality of this article. We have carefully read and studied your comments, you will find our response to the comments point by point in the file. The comments were also adopted and the relevant changes were made in the manuscript. All made changes in manuscript are marked in red at their respective places.

Notes: All line numbers used in the response letter refer to the latest submitted manuscript.

Point 1: The manuscript is clear, with some exceptions, but is relevant for the field and it is presented in a well-structured manner.

Response 1: Thank you for your comment. We have revised the manuscript sentence by sentence, and modified at several points which were highlighted in the manuscript.

Point 2: The cited references mostly are recent publications and relevant for this paperwork. Does not include excessive number of self-citations.

Response 2: Thank you very much for your comment. We have proofread and added some new references.

Point 3: The manuscript is enough scientifically sound and is the experimental design is appropriate to test the hypothesis of its article. 

Response 3: Thank you very much for your kind comment.

Point 4: The manuscript’s results are reproducible based on the details given in the methods section.

Response 4: Thank you very much for your comment.

Point 5: The figures are appropriate, but the table are extremely extended and should be synthesized in a proper manner, because they are not easy to interpret and understand. The data are interpreted appropriately and consistently throughout the manuscript, with the mentions from above. The conclusions are consistent with the evidence and arguments presented.

Response 5: Thank you very much for your valuable comments. We have adapted the tables and footnotes to make them more likely to be understand. But we have not deleted any tables because we believed each table played its own role in this manuscript.

Point 6: The ethics statements and data availability statements are enough adequate.

Response 6: Thank you for your kind appreciation.

Point 7: The nutritive evaluation system based on amino acid profiles to evaluate, compare, and rank the fresh sweet-waxy corn planted in different regions, is not enough clearly described. Why you, the authors, use it? Which was the reason due to you choose it? So, it needs more explanations from you.

Response 7: Thank you very much for your kind guidance. We have adapted and added the relevant content in right places.

Round 2

Reviewer 2 Report

Congratulations! From my point of view your paperwork could be publish now, in the present form, with some corrections (please find it in the 2 comments)
